

# Gaining insights on anyon condensation and 1-form symmetry breaking across a topological phase transition in a deformed toric code model

Joe Huxford[1*], Dung Xuan Nguyen[2†] and Yong Baek Kim[1‡]

**1** Department of Physics, University of Toronto, Ontario M5S 1A7, Canada
**2** Center for Theoretical Physics of Complex Systems, Institute for Basic Science (IBS), Daejeon 34126, Republic of Korea

★ joe.huxford@utoronto.ca , † dungmuop@gmail.com , ‡ yongbaek.kim@utoronto.ca

## Abstract

We examine the condensation and confinement mechanisms exhibited by a deformed toric code model proposed in [Castelnovo and Chamon, *Phys. Rev. B* 77, 054433 (2008)]. The model describes both sides of a phase transition from a topological phase to a trivial phase. Our findings reveal an unconventional confinement mechanism that governs the behavior of the toric code excitations within the trivial phase. Specifically, the confined magnetic charge can still be displaced without any energy cost, albeit only via the application of non-unitary operators that reduce the norm of the state. This peculiar phenomenon can be attributed to a previously known feature of the model: It maintains the non-trivial ground state degeneracy of the toric code throughout the transition. We describe how this degeneracy arises in both phases in terms of spontaneous symmetry breaking of a generalized (1-form) symmetry and explain why such symmetry breaking is compatible with the trivial phase. The present study implies the existence of subtle considerations that must be addressed in the context of recently posited connections between topological phases and broken higher-form symmetries.

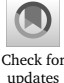

# 1 Introduction

The study of different phases of matter, and the transitions between them, is a cornerstone of condensed matter physics. Over the past few decades, we have learned much about the landscape of phases existing at zero temperature, including so-called topological phases of matter [1–3]. These topological phases, which include fractional quantum Hall systems [4–6], are characterised by long-ranged entanglement between their local degrees of freedom, which cannot be removed by local unitary evolution over a finite time [3, 7, 8]. Because of this, topological phases possess a ground state degeneracy that depends on the topology of the manifold on which they reside and which is robust to local perturbations [9, 10]. In addition, the long-ranged entanglement allows topological phases to support exotic excitations with novel exchange statistics. In 2+1d, these quasiparticles are known as anyons and have braiding statistics that generalise those of bosons and fermions. Topological phases have been a subject of intense research due to their potential applications in quantum computing and memory [9, 11–15]. Because of these applications, in addition to the study of topological phases in nature, there have been significant efforts to artificially construct topologically ordered states [16, 17].

Transitions between different topological phases are also of significant interest and illustrate how the unique properties of these phases can change and emerge. A particular class of transitions, known as condensation-confinement transitions, intimately involve the exotic excitations [18–22]. During a transition from one topological phase to another, or a transition to a topologically trivial phase, some of the anyons of the original phase may condense, meaning that their associated conserved charge is absorbed into the ground state, while other anyons become confined, unable to move without dragging an energetically costly string [18, 19]. Such transitions may involve no local order parameter, instead being described by non-local, ribbon-like quantities.

In recent years, a new perspective on topological phase transitions has emerged, building on the established Landau picture of symmetry-breaking phase transitions. It has been proposed that, instead of conventional symmetries, higher-form symmetries may be a useful descriptor of topological phases [23–27]. Unlike ordinary global symmetries, which act on an entire slice of space-time, higher-form symmetries act on lower-dimensional manifolds, of codimension two or greater in space-time. Furthermore, higher-form symmetries are topological, in the sense that their action is unaffected by smoothly deforming the manifold on which they are applied. Because of this lower dimensionality and topological nature, the charge for

a higher-form symmetry is carried by extended objects, rather than local ones as for ordinary (0-form) charge.

As an illustration of the connection between topological phases and 1-form symmetries, consider the toric code, the simplest example of a non-trivial topological order. The logical operators of the toric code are strings of $\sigma^x$ and $\sigma^z$ operators around the handles of the torus, and they are the non-trivial 1-form symmetry operators. These operators commute with the Hamiltonian and act on a lower-dimensional manifold. Furthermore, the 1-form symmetry operators are topological, meaning that we can deform the manifold on which they are applied without affecting their action, at least in the ground state manifold. These conditions define a higher-form (in this case, 1-form) symmetry. However, these 1-form symmetries are broken in the ground state manifold, as evidenced by the logical operators moving us from one ground state to another. In fact, the order parameter (charged operator) for the Wilson loop 1-form symmetry (the string of $\sigma^z$ operators about one handle) is a 't Hooft loop 1-form symmetry (a string of $\sigma^x$ about the other handle) and vice versa. This perspective on the toric code can be extended to other topological phases, where the logical operators that change the ground state by braiding anyons around a handle of the torus are non-trivial 1-form symmetries, and the braiding relations between anyons allow different logical operators to act as charged operators for each other. Together, these operators describe a pattern of spontaneous 1-form symmetry breaking in the ground state manifold [27].

Although the framework that utilizes higher-form symmetries for analyzing topological phases is still in the developmental stage, it holds significant promise for drawing upon established concepts from traditional phase transitions to enhance our comprehension of topological phases. However, to fully leverage these ideas, it is crucial to establish, in a rigorous manner, the extent to which topological phases can be described as higher-form symmetry-breaking phases and how this type of symmetry-breaking differs from that observed for ordinary (0-form) symmetries. Therefore, a careful investigation and thorough characterization of the relationship between topological phases and higher-form symmetries are essential for understanding of the topological phases of matter.

Our study presents potential complications to the currently established notions of condensation and confinement, as well as to the 1-form symmetry perspective of topological phases. To achieve this, we investigate the deformed toric code model proposed in [28], which provides exact expressions for the ground states across a phase transition. In Section 3.1, we demonstrate that the confinement of anyons may not always be enforced energetically, as non-unitary operators may exist that can transport the confined charge without dragging an energetic string. However, this comes at the cost of reducing the norm of the state, such that the norm becomes zero as the length of the transporting operator approaches infinity. This observation aligns with a view about confinement employed when studying topological phases via tensor networks, as reported in prior studies [29–32], where the ground state is studied directly, without a Hamiltonian.

In addition, as described in Section 3.2, the 't Hooft loop satisfies a perimeter law even beyond the phase transition, meaning that its expectation value $\langle T(c) \rangle$ in the ground state decays as the length $L(c)$ of the loop:

$$\langle T(c) \rangle \sim e^{-aL(c)}.$$

This is despite the apparent condensation of electric charge and confinement of magnetic charge, which we might expect to result in an area law, for which the expectation would decay with the area $A(c)$ enclosed by the loop [33]:

$$\langle T(c) \rangle \sim e^{-bA(c)}.$$

The perimeter law for the 't Hooft loop also indicates the spontaneous symmetry breaking of a 1-form symmetry [27] (which is preserved exactly by the Hamiltonian). This is analogous

to the long-ranged correlation of an order parameter for a regular symmetry breaking state, while the area law is analogous to the decay of a correlator in the disordered phase [27]. In Section 4, we discuss the connection between spontaneous symmetry breaking and topological phases in more detail, pointing out why 1-form symmetry breaking can lead to indistinguishable ground states. The perimeter law for the 't Hooft loop persists beyond the topological phase, however, indicating that spontaneous symmetry breaking of the 1-form symmetry does not always give a topological phase. We explain why the symmetry breaking alone does not guarantee indistinguishability, and what additional conditions do ensure this property.

## 2   The deformed toric code

The deformed toric code model is an exactly solvable model, based on Kitaev's toric code [9], that was introduced by Castelnovo and Chamon in Ref. [28] and further studied in Refs. [34–36] (with a related model discussed previously in Ref. [37]). By tuning a parameter in the Hamiltonian, the model can describe either a topological phase (the toric code phase) or a trivial one. This model is conveniently defined on a square lattice (although more general graphs can also be employed) with $\mathbb{Z}_2$ variables on each edge, just like the regular toric code. We will generally consider periodic boundary conditions, so that the lattice represents a toroidal manifold. Similar to the toric code, the $\mathbb{Z}_2$ degrees of freedom interact through energy terms at each vertex and plaquette, as shown in Figure 1. The vertex terms are given by

$$Q_v(\beta) = e^{-\beta \sum_{i \in \text{star}(v)} \sigma_i^z} - \prod_{i \in \text{star}(v)} \sigma_i^x . \tag{1}$$

Here $\beta$ is the previously mentioned parameter which carries the model across a phase transition, from a topologically ordered phase at low $\beta$ to a topologically trivial one at high $\beta$. At $\beta = 0$, the vertex terms are equivalent to those from the toric code (albeit with a constant shift and rescaling from their usual presentation). At small $\beta$, an expansion of the exponential indicates that the vertex term is equivalent to the toric code term plus an interaction with an applied magnetic field along the $z$-direction [28]. At larger $\beta$, however, the term differs significantly from the linear version. The $\sigma^x$ part of the term, which is the same for all $\beta$, has the effect of flipping the $\mathbb{Z}_2$ variables surrounding the vertex. Similar to the toric code, the overall effect of $Q_v(\beta)$ is to ensure that the ground state contains all configurations related by the vertex flips, although the exponential part means that this is done with a weight that depends on the number of down spins.

On the other hand, the plaquette terms are the same as those used in the toric code:

$$B_p = \prod_{i \in p} \sigma_i^z . \tag{2}$$

This term has the effect of enforcing a "no-flux" condition on the low energy space, so that the product of $\mathbb{Z}_2$ variables around each plaquette is $+1$.

The overall Hamiltonian is then given by

$$H(\beta) = - \sum_{\text{plaquettes, } p} B_p + \sum_{\text{vertices, } v} Q_v(\beta) , \tag{3}$$

so that at $\beta = 0$ the model is equivalent to the toric code.

Unlike for the regular toric code, at finite $\beta$ the energy terms are not all commuting projectors: the $Q_v(\beta)$ terms do not commute with each-other and are not projectors. Nonetheless, the lowest eigenvalue of each $Q_v(\beta)$ is zero and the ground states minimise each $Q_v(\beta)$ individually. As described in Ref. [28] and as we show in Appendix D for a more general version

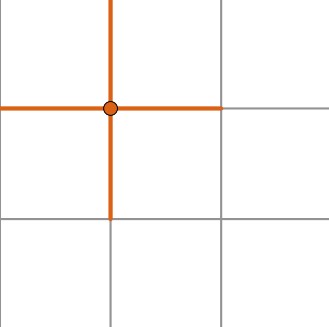
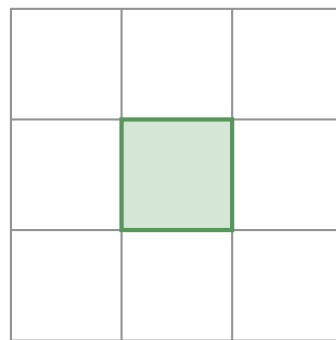

Figure 1: In the deformed toric code, the edges of the lattice host $\mathbb{Z}_2$ degrees of freedom. The vertex term (left) acts on the edges adjacent to the vertex, while the plaquette term (right) acts on the edges on the boundary of the plaquette.

of the model, this leads to the ground states taking the form

$$|GS(\beta)\rangle \propto \prod_i e^{\beta\sigma_i^z/2} |GS(0)\rangle \,, \tag{4}$$

where $|GS(0)\rangle$ is any toric code ground state. Note that this means that the ground state degeneracy of the toric code (which is four on the torus) is preserved for all values of $\beta$, even as we transition to the topologically trivial phase at large $\beta$ (although the gap closes at the critical point [38]). The product of exponentials given in Equation (4) weights the different spin configurations according to the total number of down spins. Configurations with more down spins ($\sigma_i^z = -1$) are given a smaller weighting in the ground state, while configurations with more up spins will have a greater weighting. The different ground states can be labelled by the product of spins around the handles of the torus, a fact which is maintained for any value of $\beta$ [39]. Because the spins are $\mathbb{Z}_2$ variables, this product is a parity and is given by $\pm 1$. At the extreme limit of $\beta \to \infty$, the toric code ground state with even parity about both handles becomes the fully polarized state: a simple product state of up spins [39], because the even-even parity sector already includes that polarized configuration and it is given the greatest weight as $\beta \to \infty$. On the other hand, the other ground states have odd parity around at least one handle and so must have some down spins. This means that, as $\beta \to \infty$, the ground state tends towards the configuration of spins that has fewest down spins while still satisfying the parity condition, which means having a number of down spins comparable to the linear extent of the lattice [39]. The weighting factor from Equation (4) will appear in many future expressions, so following notation from Ref. [39], we define

$$S(\beta) = \prod_i e^{\beta\sigma_i^z/2} \,. \tag{5}$$

As mentioned previously, there is a phase transition in this model as we go from low to high $\beta$. Evidence for this was given in Ref. [28], where the behaviour of the topological entanglement entropy was studied. It was found that the topological entanglement entropy is given by $S_{\text{topo.}} = \ln 2$ below a critical value of $\beta$ ($\beta_c = \frac{1}{2}\ln(1+\sqrt{2})$), but drops abruptly to zero above this critical value. This implies that the model has a transition from a topological phase (with non-zero topological entanglement entropy) to a trivial one. Further evidence of this transition was given in Ref. [39], where it was demonstrated that the magnetic susceptibility of the ground states diverges at the critical point. Furthermore, above the phase transition the magnetizations of the different degenerate ground states differ by an amount proportional to the linear extent of the lattice, indicating that the ground states are distinguishable above the

transition. This contrasts with the situation in the topological phase, where no local operators can distinguish between the ground states [2,9] (up to corrections that are exponentially small in system size), meaning that the magnetization is the same for each ground state away from the critical point.

While we will generally consider the Hamiltonian given in Equation (3), it is also possible to generalise this Hamiltonian by allowing the variable $\beta$ to vary over space or by modifying the plaquette terms in addition to the vertex terms. Despite this inhomogeneity, the ground states can still be constructed exactly. A detailed examination of this inhomogeneous model is beyond the scope of this paper, but we briefly discuss some of its properties in Appendix D.

# 3   Condensation and confinement

One way in which topological phases can undergo a phase transition to a trivial phase, or another topological one, is for some of the anyonic excitations to undergo a process called condensation. During this process, the conserved topological charge carried by the excitation is absorbed into the ground state and the excitations become trivial (often disappearing entirely). Furthermore, excitations which had non-trivial braiding with these condensing excitations in the topological phase are expected to become confined in the new phase, unable to move without dragging an energetically costly string [18]. This is because these confined excitations disturb the condensate as they move, with the string representing a location where the topological symmetries of the original ground state is restored [18]. In this work, we will examine the condensation and confinement in the deformed toric code from a variety of perspectives.

## 3.1   Ribbon operators

The first way in which we consider the pattern of condensation and confinement is to examine what happens to the original ribbon operators of the toric code as $\beta$ is increased. The toric code has two basic ribbon operators, electric and magnetic, which can be combined to give all of the different excitations in the model [9]. The electric ribbon operator (illustrated in the left side of Figure 2) is a string of $\sigma_i^z$ operators along a path in the lattice and produces vertex excitations at the two ends of the path:

$$L(t) = \prod_{i \in t} \sigma_i^z . \tag{6}$$

On the other hand, the magnetic ribbon operator (shown in the right side of Figure 2) is a string of $\sigma_i^x$ operators along a dual path (i.e., a path from plaquette to plaquette that bisects the edges of the lattice) and excites the plaquette terms at the two ends of the dual path:

$$C(s) = \prod_{j \in s} \sigma_j^x . \tag{7}$$

Both of these ribbon operators have an additional property when acting on the ground state, or an unexcited region of the lattice. They are topological, meaning that the result of acting with the ribbon operator on the ground state is unchanged if we deform the ribbon into a homotopic one (keeping the end-points fixed). This is a key property of the creation operators for anyons and allows the anyons to have braiding relations that are insensitive to local details.

Now consider what happens to these ribbon operators at finite $\beta$. The change to the Hamiltonian means that the commutation relations between the ribbon operators and energy terms

 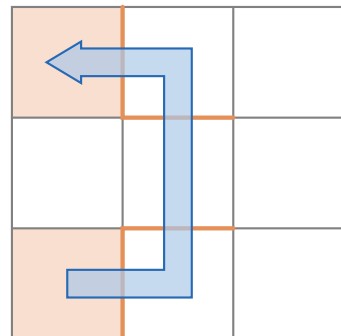

Figure 2: The electric ribbon operator (left) is a string of $\sigma^z$ operators acting along a path in the direct lattice. The magnetic ribbon operator (right) is a string of $\sigma^x$ operators acting on the edges cut by a dual path.

are altered, which may change the pattern of excitations produced by the ribbon operators. First consider the electric ribbon operator. Because the plaquette terms are unaltered as $\beta$ is increased, the electric ribbon operator still commutes with the plaquette terms (due to being a product of $\sigma^z$ operators) and does not produce any plaquette excitations. It also commutes with the exponential terms in $Q_v(\beta)$, meaning that it still commutes with vertex terms away from the ends of the ribbon (because it also commutes with $\prod_{i\in\text{star}(v)}\sigma_i^x$ there). However, at the two ends of the ribbon, the commutation relation with the $Q_v(\beta)$ terms is altered despite commuting with the exponential terms. This is because the ribbon operator anti-commutes with the product of $\sigma_i^x$ and commutes with the exponential term, leading to the following commutation relation:

$$
\begin{aligned}
Q_v(\beta)L(t)|GS(\beta)\rangle &= \left(e^{-\beta\sum_{i\in\text{star}(v)}\sigma_i^z} - \prod_{i\in\text{star}(v)}\sigma_i^x\right)L(t)|GS(\beta)\rangle \\
&= L(t)\left(e^{-\beta\sum_{i\in\text{star}(v)}\sigma_i^z} + \prod_{i\in\text{star}(v)}\sigma_i^x\right)|GS(\beta)\rangle \\
&= L(t)\left(2e^{-\beta\sum_{i\in\text{star}(v)}\sigma_i^z} - Q_v(\beta)\right)|GS(\beta)\rangle \ .
\end{aligned}
\tag{8}
$$

In the ground state space, $Q_v(\beta)|GS(\beta)\rangle = 0$ and so

$$
Q_v(\beta)L(t)|GS(\beta)\rangle = L(t)2e^{-\beta\sum_{i\in\text{star}(v)}\sigma_i^z}|GS(\beta)\rangle \ .
\tag{9}
$$

The state $L(t)|GS(\beta)\rangle$ is not generally an eigenstate of $Q_v(\beta)$ or an energy eigenstate. In fact, it has an increasing overlap with the ground state as $\beta$ increases, which we can see from the fact that the magnetization is non-zero above the critical $\beta$ [39], so the $\sigma_i^z$ that make up the ribbon operator acquire a non-zero expectation value. More precisely, the expectation value of the ribbon operator can be mapped to a spin-spin correlation in the classical Ising model [28], as we discuss further in the Supplemental Material, which is long-ranged above the critical value of $\beta$ and tends towards 1. This increasing expectation value indicates that the electric excitations of the toric code join the ground state as $\beta$ is increased, demonstrating that they condense during the phase transition.

Next consider the magnetic ribbon operators. Once again, the plaquette terms are unchanged as $\beta$ is increased and so their commutation relations with the ribbon operator are unaltered. Specifically, the ribbon operator commutes with the plaquette terms away from the ends of the ribbon and anti-commutes with the plaquette terms at the two ends. On the other hand, the commutation relations with the vertex terms are significantly different for finite $\beta$.

The ribbon operator does not commute with the exponential terms in $Q_v(\beta)$ for any vertices adjacent to the ribbon, leading to the following commutation relations for such vertices:

$$
\begin{aligned}
Q_v(\beta)C(t)|GS(\beta)\rangle &= \left(e^{-\beta\sum_{i\in\text{star}(v)}\sigma_i^z} - \prod_{i\in\text{star}(v)}\sigma_i^x\right)C(t)|GS(\beta)\rangle \\
&= C(t)\left(e^{-\beta\sum_{i\in\text{star}(v)}\sigma_i^z}e^{2\beta\sum_{i\in\text{star}(v)\cap t}\sigma_i^z} - \prod_{i\in\text{star}(v)}\sigma_i^x\right)|GS(\beta)\rangle \\
&= C(t)\left(e^{-\beta\sum_{i\in\text{star}(v)}\sigma_i^z}(e^{2\beta\sum_{i\in\text{star}(v)\cap t}\sigma_i^z}-1) + Q_v(\beta)\right)|GS(\beta)\rangle \\
&= C(t)e^{-\beta\sum_{i\in\text{star}(v)}\sigma_i^z}(e^{2\beta\sum_{i\in\text{star}(v)\cap t}\sigma_i^z}-1)|GS(\beta)\rangle\,.
\end{aligned}
\tag{10}
$$

This non-commutation with the adjacent vertex terms indicates that the magnetic ribbon operator may produce excitations along its length, suggesting that the magnetic charge is confined.

Despite this, however, there is a way to define a ribbon operator that moves magnetic charge but which has no energy cost associated to its length. Consider the non-unitary operator

$$
\tilde{C}(t) = S(\beta)C(t)S(\beta)^{-1}\,,
\tag{11}
$$

which we will call the deformed magnetic ribbon operator. Note that the form of this operator implies that it locally removes the condensate, applies the ordinary magnetic ribbon operator and then replaces the condensate. This deformed magnetic ribbon operator has the following commutation relations with the vertex terms:

$$
Q_v(\beta)\tilde{C}(t) = \tilde{C}(t)\left(\prod_{j\in t\cap\text{star}(v)}e^{2\beta\sigma_j^z}\right)Q_v(\beta)\,.
\tag{12}
$$

While the deformed magnetic ribbon operator does not commute with the vertex terms, it does satisfy the relation

$$
Q_v(\beta)\tilde{C}(t)|GS(\beta)\rangle = \tilde{C}(t)\left(\prod_{j\in t\cap\text{star}(v)}e^{2\beta\sigma_j^z}\right)Q_v(\beta)|GS(\beta)\rangle = 0\,,
\tag{13}
$$

where we used the fact that $Q_v(\beta)$ annihilates the ground state space in the last step. This indicates that the deformed ribbon operator does not produce any vertex excitations along its length when acting on the ground state. Because the deformed ribbon operator does still move magnetic charge, this implies that it is possible to move magnetic charge without producing an energetically costly linking string. However, this conflicts with the standard intuition that an excitation with non-trivial braiding relations with a condensing charge should become energetically confined.

One resolution to this comes from the non-unitary nature of the deformed ribbon operator and a connection to a tensor network description of topological phases. In Ref. [29], it is described how anyons can be created from a tensor network description of a topological ground state by applying so-called matrix product operators (MPOs). These operators act on the virtual layer of the tensor network, modifying the nearby tensors. Analogous to the topological property of ribbon operators, the MPOs possess a "pull-through" property, meaning that they can be deformed in the virtual layer without affecting the state. This property is guaranteed by a virtual symmetry possessed by the tensors. Significantly, it is possible to alter the ground state while preserving the virtual symmetry and the MPOs, even across a phase transition. In this case, the confinement of an anyon type can be diagnosed from its MPO. If the norm of the state with the MPO inserted decays exponentially with its length (represented by a sub-unity eigenvalue for the relevant transfer matrix), the anyon is said to be confined, because it is not possible to produce an isolated anyon by inserting a semi-infinite MPO string. As the length of the string tends to infinity (in the thermodynamic limit), the norm of the state becomes zero.

This norm-based picture of confinement would seem to apply equally well to the non-unitary (and therefore norm-nonconserving) ribbon operator we described earlier. In fact, by constructing a tensor network description of the deformed toric code, we verified that an MPO on the virtual layer lifts to a deformed magnetic ribbon operator on the real layer, as we describe in more detail in Appendix A. The deformed magnetic ribbon operator is topological when applied on the ground state, reflecting the pull-through property of the MPO.

## 3.2 Topological charge measurement: 't Hooft and Wilson loops

In addition to creating and moving the basic excitations, the toric code ribbon operators can be used to study the behaviour of the conserved topological charge across the transition. This is because the closed ribbon operators can be used to construct topological charge measurement operators, which project onto states of definite topological charge [40]. In the case of the toric code, these measurement operators are equivalent to the Wilson loops (which measure magnetic charge) and the 't Hooft loops (which measure electric charge), up to additive and multiplicative shifts. The Wilson loop is a closed electric ribbon operator:

$$W(c) = \prod_{i \in c} \sigma_i^z \,, \tag{14}$$

while the 't Hooft loop is a closed magnetic ribbon operator:

$$T(c) = \prod_{i \in c} \sigma_i^x \,. \tag{15}$$

These operators have eigenvalues of 1 for states in which they enclose no magnetic charge (for the Wilson loop) or electric charge (for the 't Hooft loop). On the other hand, they have eigenvalues of $-1$ for states in which they enclose non-trivial magnetic or electric charge respectively (such as a state in which they enclose a single excitation).

In the toric code ground state space, contractible Wilson and 't Hooft loops both have a constant expectation value of 1, independent of the length or area of the loop. This indicates that the ground state possesses no magnetic or electric charge. As we deform away from the toric code fixed point, one or both of these operators are expected to decay with the size of the loop, either according to its length (which is called perimeter law) or its area (which is called area law), implying the presence of some non-trivial charge in the ground state. For the deformed toric code, we find that the Wilson loop satisfies a zero law $\langle W(c) \rangle = 1$, which reflects the fact that the exponential term in the Hamiltonian only includes $\sigma^z$ operators, so that there is absolutely no mixing of magnetic excitations into the ground state (even a small amount of mixing would be expected to produce a perimeter law). On the other hand, the 't Hooft loop satisfies a perimeter law for all non-zero values of $\beta$, as we show in Appendix C. By contrast, if we had used a linear coupling to a magnetic field along $z$ rather than the exponential term of the deformed toric code to energetically punish down spins, the 't Hooft loop would have decayed with an area law [41], which is typically an indicator of confinement of magnetic charge and condensation of electric charge (the Wilson loop would still satisfy a zero law). It is possible that this perimeter law for the deformed toric code model reflects the unusual (non-energetic) confinement of the magnetic charge which we discussed in Section 3.1. A perimeter law can also arise from the application of a magnetic field at an angle away from the $z$ axis, but then we would also expect the Wilson loop to satisfy a perimeter law rather than a zero law [41].

The deformed magnetic ribbon operator can also be used to define a "deformed 't Hooft loop":

$$\tilde{T}(c) = \prod_{i \in c} e^{\beta \sigma_i^z / 2} \sigma_i^x e^{-\beta \sigma_i^z / 2} \,. \tag{16}$$

Similar to the Wilson loop, this has a zero-law expectation value in the ground state for a contractible loop (indeed, the deformed 't Hooft loop is topological in the ground state). However, the interpretation of this operator is less clear. Rather than just measure electric charge like the 't Hooft loop, it also moves it around due to the exponential terms, and can be considered as a 't Hooft loop dressed with electric ribbon operators.

While we have so far considered contractible loop operators, which can be used to diagnose charge condensation in a ground state, we can also construct non-contractible loop operators which wrap around one or both of the handles of the torus. As we will see in the next section, these have important consequences for the ground state degeneracy of the deformed toric code.

# 4 Ground state properties and 1-form symmetry

## 4.1 Ground state structure

An interesting feature of the deformed toric code model is that it has the same non-trivial ground state degeneracy above and below the phase transition [28, 39]. This is unusual because the degeneracy of the toric code, which depends on the manifold on which it is placed, is tied to the topological nature of the phase and would typically be destroyed after a phase transition to a trivial phase. To understand this ground state degeneracy, we note that a convenient basis for the ground state space is described by states of definite parity, which are eigenstates of the Wilson loop operators around the two handles of the torus. That is, given two Wilson loop operators

$$W(c_1) = \prod_{i \in c_1} \sigma_i^z \,,$$
$$W(c_2) = \prod_{i \in c_2} \sigma_i^z \,,$$

where $c_1$ and $c_2$ are closed paths around the two handles of the torus (the precise choice of path does not matter due to the topological nature of the Wilson loops), we can define a basis of ground states

$$\{ \, |GS_{++}(\beta)\rangle, \, |GS_{+-}(\beta)\rangle, \, |GS_{-+}(\beta)\rangle, \, |GS_{--}(\beta)\rangle \, \} \,,$$

by

$$W(c_1)|GS_{\lambda_1\lambda_2}(\beta)\rangle = \lambda_1 |GS_{\lambda_1\lambda_2}(\beta)\rangle \,, \tag{17}$$
$$W(c_2)|GS_{\lambda_1\lambda_2}(\beta)\rangle = \lambda_2 |GS_{\lambda_1\lambda_2}(\beta)\rangle \,. \tag{18}$$

This basis is similar to one used for the regular toric code [9] (corresponding to $\beta = 0$) and follows from the relationship between toric code ground states and deformed toric code ground states:

$$|GS_{\lambda_1\lambda_2}(\beta)\rangle = \frac{1}{\sqrt{N_{\lambda_1\lambda_2}(\beta)}} S(\beta)|GS_{\lambda_1\lambda_2}(0)\rangle \,, \tag{19}$$

where $S(\beta)$ commutes with the Wilson loops and therefore preserves their eigenvalues. Here $N_{\lambda_1\lambda_2}(\beta)$ is a normalization factor that generally varies between ground states.

In the case of the toric code, these basis ground states are related by the actions of 't Hooft loop operators around the handles of the torus. Indeed, the ground state degeneracy is a result of the anticommutation relation between the Wilson and 't Hooft loops [9]. For

finite $\beta$, however, the 't Hooft loop does not commute with the Hamiltonian and does not move us between these different ground states. Instead, that role is played by the deformed 't Hooft loop defined in Equation (16), which still does not commute with the Hamiltonian either but does preserve the ground state space. These deformed 't Hooft loops obey the same anticommutation relations with the Wilson loops as the regular 't Hooft loops do in the toric code. Namely, the deformed 't Hooft loop $\tilde{T}(\bar{c}_2)$ applied on a dual path around one handle anticommutes with the Wilson loop $W(c_1)$ applied around the other handle, but commutes with the Wilson loop applied on the same handle:

$$W(c_1)\tilde{T}(\bar{c}_2) = -\tilde{T}(\bar{c}_2)W(c_1), \tag{20}$$

$$W(c_1)\tilde{T}(\bar{c}_1) = +\tilde{T}(\bar{c}_1)W(c_1). \tag{21}$$

This anticommutation relation means that applying the deformed 't Hooft loop to a ground state $|GS_{\lambda_1\lambda_2}(\beta)\rangle$ changes its eigenvalue with respect one of the Wilson loop operators:

$$W(c_1)\tilde{T}(\bar{c}_2)|GS_{\lambda_1\lambda_2}(\beta)\rangle = -\tilde{T}(\bar{c}_2)W(c_1)|GS_{\lambda_1\lambda_2}(\beta)\rangle$$
$$= -\lambda_1\tilde{T}(\bar{c}_2)|GS_{\lambda_1\lambda_2}(\beta)\rangle, \tag{22}$$

$$W(c_2)\tilde{T}(\bar{c}_2)|GS_{\lambda_1\lambda_2}(\beta)\rangle = \tilde{T}(\bar{c}_2)W(c_2)|GS_{\lambda_1\lambda_2}(\beta)\rangle$$
$$= \lambda_2\tilde{T}(\bar{c}_2)|GS_{\lambda_1\lambda_2}(\beta)\rangle. \tag{23}$$

Because the deformed 't Hooft loop preserves the ground state space, this means that

$$\tilde{T}(\bar{c}_2)|GS_{\lambda_1\lambda_2}(\beta)\rangle \propto |GS_{-\lambda_1\lambda_2}(\beta)\rangle.$$

More precisely, by using the fact that the regular 't Hooft loop is unitary and so the constant of proportionality is one (up to a phase) for the $\beta = 0$ case, we have

$$\tilde{T}(\bar{c}_2)|GS_{\lambda_1\lambda_2}(\beta)\rangle = S(\beta)T(\bar{c}_2)S(\beta)^{-1}|GS_{\lambda_1\lambda_2}(\beta)\rangle$$
$$= S(\beta)T(\bar{c}_2)S(\beta)^{-1}\frac{1}{\sqrt{N_{\lambda_1\lambda_2}(\beta)}}S(\beta)|GS_{\lambda_1\lambda_2}(0)\rangle$$
$$= \frac{1}{\sqrt{N_{\lambda_1\lambda_2}(\beta)}}S(\beta)T(\bar{c}_2)|GS_{\lambda_1\lambda_2}(0)\rangle$$
$$= \sqrt{\frac{N_{-\lambda_1\lambda_2}(\beta)}{N_{\lambda_1\lambda_2}(\beta)}}|GS_{-\lambda_1\lambda_2}(\beta)\rangle. \tag{24}$$

A similar result holds for $\tilde{T}(\bar{c}_1)$, except that it introduces a minus sign to the other eigenvalue. We see that the deformed 't Hooft loop operators move us between the basis ground states, but with a potential normalization factor because the deformed 't Hooft loops are not unitary. While we have started with the existence of this basis, the anticommutation can also be used to argue for that basis in the first place. Because the Wilson loop operators commute with the Hamiltonian and each-other, we can simultaneously diagonalize them, meaning that we always have a ground state labelled by some pair of eigenvalues $\lambda_1, \lambda_2$. By applying the deformed 't Hooft loops, we then generate ground states labelled by all the other possible pairs of eigenvalues, which gives us fourfold degenerate ground states.

So far, this reasoning is identical to that for the regular toric code [9], except for the presence of normalization factors in Equation (24). For any finite system size and finite $\beta$, these normalization factors are non-zero, although they may tend to zero or infinity in the thermodynamic limit. Indeed, in Appendix B, we use a mapping between the normalization of the ground states and the classical Ising partition function [28] to show that the normalization

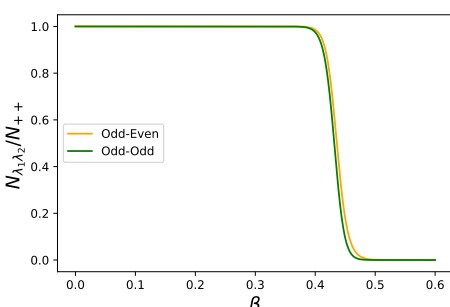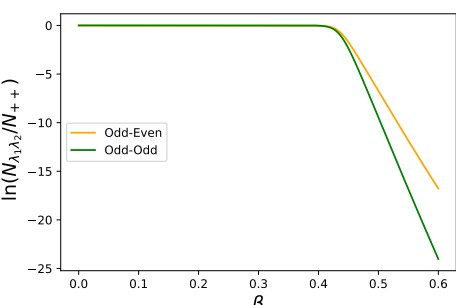

Figure 3: We plot the ratio of the odd-even and odd-odd normalization factors to the even-even normalization factor as a function of $\beta$ for a 30 by 30 system. For systems with equal horizontal and vertical dimensions the even-odd factor is the same as the odd-even factor. The left figure shows the ratio and the right figure shows the natural logarithm of the ratio. We see a transition from a constant value (of 1) below the phase transition (at $\beta_c \approx 0.44$) to exponential decay above the phase transition.

factor is approximately one in the topological phase, indicating that the deformed 't Hooft operator acts approximately unitarily in the topological phase, but rapidly drops towards zero in the trivial phase. This is illustrated in Figure 3, where we plot the ratio of the normalization factors of the odd-even ($\lambda_1 \lambda_2 = -+$) and odd-odd ($\lambda_1 \lambda_2 = --$) ground states to the factor for the even-even ($\lambda_1 \lambda_2 = ++$) ground state. Because the non-unitarity reflects the confinement of the magnetic particle, the ratio dropping to zero (or rising to infinity) in the trivial phase is associated to the impossibility of infinitely separating two confined magnetic particles in the thermodynamic limit. We may worry that, because the ratio of normalization factors goes to zero (or infinity) in the thermodynamic limit, some of the states that are related by the action of the deformed 't Hooft operator may become ill-defined in that limit. However, this does not seem to be the case as we can construct all the ground states through Equation (19), with the operator $S(\beta)$ not mixing the ground states labelled by the eigenvalues of the Wilson operators.

To summarize, the deformed toric code has the same ground state degeneracy as the regular toric code for all $\beta$, despite the model describing a trivial phase above a critical value of $\beta$ [28]. Furthermore, the ground state space admits a basis labelled by eigenvalues of the Wilson loop operators, just like for the toric code [39]. However, the operators that move us between these basis states are the non-unitary deformed 't Hooft loops, rather than the regular 't Hooft loops.

## 4.2  1-form symmetry breaking picture

This unusual ground state degeneracy in the trivial phase gains additional significance in light of recent discussions of higher-form symmetries. As explained in the introduction, a connection has recently been established between topological phases and spontaneous symmetry breaking of 1-form symmetries [26, 27]. Higher-form symmetries are generalized versions of symmetries, where the symmetry acts on a lower dimensional (closed) subspace, rather than across all of space. They possess the same type of composition law as regular symmetries, so that two higher-form symmetries applied on the same position combine via group multiplication. However, higher-form symmetries must also satisfy a topological condition, so that the manifold on which we apply a higher-form symmetry can be deformed smoothly without affecting the action of the symmetry. Like regular symmetries, we say that objects on which the higher-form symmetries act non-trivially are charged and we can decompose this action in

terms of irreps of the group that describes the composition of the higher-form symmetries on a particular manifold. However, these charged objects will generally be extended rather than local. This combination of familiar properties along with new features makes higher-form symmetries appealing to work with and potentially useful for the description of topological phases.

To understand how higher-form symmetries may be useful for topological phases, consider how higher-form symmetries appear in 2+1d lattice models. In this context, we are most interested in 1-form symmetries, which are defined on closed manifolds of one lower dimension than the spatial manifold [25], meaning closed ribbons. The charged objects for the 1-form symmetry are also extended objects and in the simplest case are also defined on ribbons. Given a 1-form symmetry $S(c)$ and charged operator $V(r)$, the two operators satisfy a commutation relation of the form [27]

$$S(c)V(r) = e^{i\phi(c,r)}V(r)S(c),\tag{25}$$

where $e^{i\phi(c,r)}$ is a phase which depends on how $r$ and $c$ intersect in the lattice and which is a representation of the 1-form symmetry group. This is completely analogous to the relationship between a 0-form symmetry and its order parameter: the order parameter transforms under a non-trivial representation of the symmetry group. Just like for a 0-form symmetry, if the expectation value of that order parameter is non-zero in a ground state, then the state spontaneously breaks the 1-form symmetry.

This picture can be immediately translated to the toric code fixed point. Indeed, the logical operators that we used to describe the ground state space in Section 4.1 are simply 1-form symmetries and order parameters. The Wilson loops are 1-form symmetries on the ground state space: they are operators that are defined on ribbons, which commute with the Hamiltonian and which are topological. The 't Hooft loops then act as order parameters for the Wilson loops, with the anti-commutation relation between them reflecting the $\mathbb{Z}_2$ nature of each 1-form symmetry (i.e., the phase $e^{i\phi(c,r)} = \pm 1$). The ground state space then contains both symmetric and symmetry-breaking states. We therefore see that the ground state degeneracy of the toric code, originating from the non-commutativity of the logical operators, can equivalently be described by 1-form symmetry breaking, with the Wilson loops acting as the 1-form symmetry. While we have treated the Wilson loops as the 1-form symmetry and the 't Hooft loops as the order parameter, we can equally reverse these roles because they both satisfy the conditions for 1-form symmetries. As we will discuss later, this duality is important for protecting one of the defining properties of topological phases and this is where the deformed toric code differs from the undeformed model.

To some extent, we can still interpret the deformed toric code in terms of 1-form symmetries. The Wilson loop operators are still topological and commute with the Hamiltonian, so they are 1-form symmetry operators. The 't Hooft loop operators are not, however, because they neither commute with the Hamiltonian nor are topological on the ground state space. The expectation value of the contractible 't Hooft loop operators does satisfy a perimeter law, which can be taken as an indicator for spontaneous breaking of the Wilson loop 1-form symmetry [27]. A clearer picture of the symmetry breaking can be obtained by considering the non-contractible deformed 't Hooft loop operators. These operators, while they are topological and preserve the ground state space, are not unitary. However, they obey the same anti-commutation relations with the Wilson loops as the 't Hooft loops and so can act as order parameters, which do not need to be unitary. While the basis ground states $|GS_{\lambda_1 \lambda_2}(\beta)\rangle$ defined in Equation (19) are symmetric under the 1-form symmetry and so have zero expectation value for the deformed 't Hooft loops, a general ground state will have some finite expectation

value:

$$\sum_{\lambda_1,\lambda_2} a^*_{\lambda_1\lambda_2} \langle GS_{\lambda_1\lambda_2}(\beta)|\tilde{T}(\bar{c}_2) \sum_{\lambda'_1,\lambda'_2} a_{\lambda'_1\lambda'_2} |GS_{\lambda'_1\lambda'_2}(\beta)\rangle$$

$$= \sum_{\lambda_1,\lambda_2} a^*_{\lambda_1\lambda_2} \langle GS_{\lambda_1\lambda_2}(\beta)| \sum_{\lambda'_1,\lambda'_2} a_{\lambda'_1\lambda'_2} \sqrt{\frac{N_{-\lambda'_1\lambda'_2}}{N_{\lambda'_1\lambda'_2}}} |GS_{-\lambda'_1\lambda'_2}(\beta)\rangle$$

$$= \sum_{\lambda_1,\lambda_2} a^*_{\lambda_1\lambda_2} \sum_{\lambda'_1,\lambda'_2} a_{\lambda'_1\lambda'_2} \sqrt{\frac{N_{-\lambda'_1\lambda'_2}}{N_{\lambda'_1\lambda'_2}}} \delta(\lambda_1,-\lambda'_1)\delta(\lambda_2,\lambda'_2)$$

$$= \sum_{\lambda_1,\lambda_2} a^*_{\lambda_1\lambda_2} \sum_{\lambda'_1,\lambda'_2} a_{-\lambda_1\lambda_2} \sqrt{\frac{N_{\lambda_1\lambda_2}}{N_{-\lambda_1\lambda_2}}} \delta(\lambda_1,-\lambda'_1)\delta(\lambda_2,\lambda'_2)$$

$$= \sum_{\lambda_1,\lambda_2} a^*_{\lambda_1\lambda_2} a_{-\lambda_1\lambda_2} \sqrt{\frac{N_{\lambda_1\lambda_2}}{N_{-\lambda_1\lambda_2}}}, \tag{26}$$

where $\delta(\lambda_2,\lambda'_2)$ is the Kronecker delta. This implies that the ground state space exhibits spontaneous symmetry breaking of the Wilson loop 1-form symmetries, which is reflected in the expectation value of the deformed 't Hooft loops. This is further indicated by the fact that general ground states are not symmetric under the Wilson loop 1-form symmetries. However, we know that above a critical value of $\beta$, the deformed toric code is in a trivial phase. Therefore we need to distinguish between degeneracy from spontaneous 1-form symmetry breaking in the ground state space and true topologically protected degeneracy.

## 4.3 Conditions for indistinguishability

To understand this distinction, we must first understand the key features of topologically protected degeneracy. Topological phases are characterised by *indistinguishable* degenerate (or nearly degenerate) ground states, which cannot be connected by any local operators. Instead, the different ground states are connected by non-local ribbon-like operators. It has been claimed [26, 27] that this is equivalent to the picture of symmetry breaking of 1-form symmetry, for which the ground states are connected by 1-form symmetry operators. The indistinguishability of the degenerate topological ground states is then a consequence of being connected by these 1-form symmetry operators. Here we aim to expand on this argument and demonstrate some caveats, illustrating these with the example of the deformed toric code model.

Firstly, we give a definition for the notion of indistinguishability, as it pertains to the ground states of topological phases. There are two key ingredients to this indistinguishability (as described in Ref. [42], for example):

1. Any local operator has the same expectation value in each ground state:

$$\langle GS\,1|\hat{O}|GS\,1\rangle = \langle GS\,2|\hat{O}|GS\,2\rangle\,, \tag{27}$$

2. Two orthogonal ground states cannot be connected by any local operator:

$$\langle GS\,1|\hat{O}|GS\,2\rangle = 0\,. \tag{28}$$

In fact, these two properties are related, in that if one of them holds for the entire ground state space then the other will too. However, when considering a particular basis of states

(such as the symmetry broken states), the two are distinct. This is important because ordinary symmetry broken states might also satisfy the second condition. For example the two fully polarised states that are the ground states of the zero field Ising model are connected by flipping every spin simultaneously, not by local operators. However, they obviously have different expectation values for the magnetization and so are distinguishable by local operators in that sense. In addition, the symmetric and antisymmetric linear combinations of the two symmetry breaking ground states, which are also ground states, can be connected by local operators by measuring the magnetization.

Now suppose that we have two ground states that are connected by a 1-form symmetry operator: $|GS\ 1\rangle$ and $|GS\ 2\rangle = S(c)|GS\ 1\rangle$. For any local operator $\hat{O}$, we have

$$\langle GS\ 2|\hat{O}|GS\ 2\rangle = \langle GS\ 1|S(c)^\dagger \hat{O} S(c)|GS\ 1\rangle . \tag{29}$$

It is possible that the support of the local operator $\hat{O}$ intersects with the ribbon $c$, and so may fail to commute with the 1-form symmetry operator. However, the 1-form symmetry operator is topological on the ground state space, and so may freely be deformed to a position $c'$ such that it does not intersect with $\hat{O}$. Then we have

$$\begin{aligned}\langle GS\ 2|\hat{O}|GS\ 2\rangle &= \langle GS\ 1|S(c')^\dagger \hat{O} S(c')|GS\ 1\rangle \\ &= \langle GS\ 1|S(c')^\dagger S(c') \hat{O}|GS\ 1\rangle \\ &= \langle GS\ 1|\hat{O}|GS\ 1\rangle . \end{aligned} \tag{30}$$

We therefore see that spontaneous symmetry-breaking of the 1-form symmetry guarantees that the symmetry-broken states satisfy the first condition for indistinguishability: they have the same expectation value for any local operator. However, the spontaneous symmetry breaking is not enough to guarantee the second condition. Furthermore, it only guarantees the first condition on the symmetry-broken states, not a linear combination of them (such as the symmetric states) and so need not hold for the entire ground state space.

Now suppose that there is a second 1-form symmetry $\tilde{S}(r)$ which acts as an order parameter for the first, in the sense that the symmetry-broken states are eigenstates of the second 1-form symmetry with different eigenvalues:

$$\tilde{S}(r)|GS\ 1\rangle = e^{i\theta_1}|GS\ 1\rangle ,$$
$$\tilde{S}(r)|GS\ 2\rangle = e^{i\theta_2}|GS\ 2\rangle .$$

Then for a local operator $\hat{O}$, the matrix element between the two ground states can be written as

$$\langle GS\ 2|\hat{O}|GS\ 1\rangle = e^{i(\theta_2-\theta_1)}\langle GS\ 2|S^\dagger(r)\hat{O}S(r)|GS\ 1\rangle ,$$

where we can choose the position of $r$ to ensure that $S(r)$ does not overlap with $\hat{O}$ and so commutes with it. Then

$$\begin{aligned}\langle GS\ 2|\hat{O}|GS\ 1\rangle &= e^{i(\theta_2-\theta_1)}\langle GS\ 2|S^\dagger(r)S(r)\hat{O}|GS\ 1\rangle \\ &= e^{i(\theta_2-\theta_1)}\langle GS\ 2|\hat{O}|GS\ 1\rangle , \end{aligned} \tag{31}$$

where the last line follows from unitarity of the 1-form symmetry. Given that the two eigenvalues are different, this equality can only hold if

$$\langle GS\ 2|\hat{O}|GS\ 1\rangle = 0, \tag{32}$$

from which we see that having a second 1-form symmetry as an order parameter guarantees that the symmetry-broken states also satisfy the second ingredient of indistinguishability. Note

that to obtain this result, we had to use the topological nature of the 1-form symmetry order parameter in addition to its unitarity.

With the second 1-form symmetry guaranteeing that both indistinguishability properties are satisfied by the symmetry-broken states, the same properties are guaranteed for any linear combination of these states. For example, an arbitrary combination $\alpha \,|GS\ 1\rangle + \beta \,|GS\ 2\rangle$ satisfies

$$
\begin{aligned}
(\alpha^* \langle GS\ 1| + \beta^* \langle GS\ 2|)\hat{O}(\alpha \,|GS\ 1\rangle + \beta \,|GS\ 2\rangle) = |\alpha|^2 \langle GS\ 1|\hat{O}\,|GS\ 1\rangle + |\beta|^2 \langle GS\ 1|\hat{O}\,|GS\ 1\rangle \\
+ \alpha^*\beta \langle GS\ 1|\hat{O}\,|GS\ 2\rangle + \beta^*\alpha \langle GS\ 2|\hat{O}\,|GS\ 1\rangle \,.
\end{aligned}
$$

We have already seen that the cross-terms are zero and the diagonal matrix elements are the same for each ground state, giving us

$$
(\alpha^* \langle GS\ 1| + \beta^* \langle GS\ 2|)\hat{O}(\alpha \,|GS\ 1\rangle + \beta \,|GS\ 2\rangle) = (|\alpha|^2 + |\beta|^2) \langle GS\ 1|\hat{O}\,|GS\ 1\rangle \,,
$$

which is just equal to $\langle GS\ 1|\hat{O}\,|GS\ 1\rangle$ for a normalized state. A similar result holds for the second indistinguishability property.

We have seen that indistinguishability is guaranteed by having two 1-form symmetries that act as order parameters for each-other. Simply having a spontaneously broken 1-form symmetry is not enough to guarantee indistinguishability. We can apply this idea to the toric code, which does have indistinguishable ground states. The 1-form symmetries that protect this property are the 't Hooft and Wilson loops around opposite handles, both of which are 1-form symmetries and which anti-commute with each-other. The 't Hooft loops act as the order parameters for the Wilson loops and vice-versa, meaning that the toric code satisfies the condition for indistinguishability that we laid out above. This can be thought of as a mixed 't Hooft anomaly between the two 1-form symmetries, corresponding to the non-trivial braiding of the anyons.

This idea also allows us to explain the non-trivial degeneracy that is present in the deformed toric code even above the transition to a trivial phase. As described earlier, the deformed toric code has spontaneous symmetry breaking of a 1-form symmetry on both sides of the phase transition. This symmetry breaking gives rise to the ground state degeneracy in both the topological and the trivial phases across the topological phase transition. However, the order parameter for this symmetry breaking (the deformed 't Hooft loop) is not itself a 1-form symmetry and so the symmetry breaking does not guarantee that the ground states are indistinguishable (in the anomaly language, we no longer have the mixed 't Hooft anomaly because one of the symmetries is explicitly broken). Below the phase transition, we expect the robustness of topological order to perturbations to protect the indistinguishability anyway. The 't Hooft loop 1-form symmetry should persist in some form in the low energy space as an *emergent* symmetry, which is robust [43]. Furthermore, as discussed in Section 4.1 (with proof given in Appendix B), the deformed 't Hooft loop acts in an approximately unitary way and so can be treated as an approximate 1-form symmetry in the topological phase, meaning that it protects indistinguishability. However, no such protection extends to the trivial phase. This can be verified by examining the properties of the ground states, as reported in Ref. [39]. Below the phase transition all of the ground states have approximately the same value of magnetization, but above it these values differ [39], indicating that the ground states are distinguishable. This provides a concrete example of the caveats to the connection between spontaneous breaking of 1-form symmetries and topological phases.

We should note that the non-topological spontaneous symmetry breaking, meaning symmetry breaking that does not result in indistinguishable ground states, is fragile almost by definition, unlike for the topological phase. If we add an arbitrary but infinitesimal local perturbation to the Hamiltonian, which will select the ground state with lowest expectation value for that perturbation, it will always select one of the symmetric states (if it induces any splitting

at all), rather than a symmetry breaking state. This follows from the fact that the symmetric states are not connected by any local operator, meaning that the expectation value of any local operator for any linear combination of the symmetric states only depends on the (modulus squared) of the amplitude for each symmetric state (because cross terms are zero) and will be minimised in one of the symmetric states. For example, given two symmetric states $|+\rangle$ and $|-\rangle$, we have $\langle+|\hat{O}|-\rangle = 0$ (from Equation (32)) and so the expectation value for $\hat{O}$ in the state $\alpha|+\rangle + \beta|-\rangle$ is $|\alpha|^2\langle+|\hat{O}|+\rangle + |\beta|^2\langle-|\hat{O}|-\rangle$, which is minimised by taking $\alpha = 1$ or $\beta = 1$ depending on which matrix element is smaller. This indicates that, at first order in degenerate perturbation theory, a symmetric state is always selected by any local perturbation which induces a splitting. In addition, there must always be a local operator that induces such a splitting, otherwise the states would satisfy both conditions for indistinguishability (i.e., the symmetric states would have the same expectation value for all local operators as well as not being connected by an local operators).

## 5 Application to other commuting projector models

While we have been considering a deformed version of the toric code so far, other commuting projector models admit similar deformations [30]. Suppose we start with a Hamiltonian

$$H = \sum_j h_j,$$

where the $h_j$ are local commuting projectors which can be independently satisfied and let $|\Omega\rangle$ be one of its ground states. Then for invertible and positive operators $s_i$ acting on individual degrees of freedom, the state $\left(\prod_i s_i\right)|\Omega\rangle$ is a ground state of the Hamiltonian

$$\tilde{H} = \sum_j S(j)^{-1} h_j S(j)^{-1}, \tag{33}$$

where

$$S(j) = \prod_{i\in\text{Support}(h_j)} s_i. \tag{34}$$

Because $S(j)$ is a product of Hermitian terms on different sites, $S(j)$ is Hermitian and so is $S(j)^{-1}$. This means that the energy terms $S(j)^{-1} h_j S(j)^{-1}$ are also Hermitian, as we require.

The fact that this Hamiltonian does indeed support the ground states $\left(\prod_i s_i\right)|\Omega\rangle$ for a general initial commuting projector model can be proved in the same way as Ref. [28] did for the deformed toric code model. The ground state $|\Omega\rangle$ satisfies $h_j|\Omega\rangle = \lambda_j^{\min}|\Omega\rangle$ for all $j$, because the Hamiltonian is made up of commuting local terms that can be independently satisfied. In addition, the $h_j$ are projectors, so $\lambda_j^{\min} = 0$ for all $j$, meaning that $h_j|\Omega\rangle = 0$. Even if a term $h_j$ is not a projector then we can add a constant to them so that its lowest eigenvalue is zero to obtain the same result. Then

$$S(j)^{-1} h_j S(j)^{-1}\left(\prod_i s_i\right)|\Omega\rangle = S(j)^{-1} h_j\left(\prod_{i\notin\text{Support}(h_j)} s_i\right)|\Omega\rangle$$

$$= S(j)^{-1}\left(\prod_{i\notin\text{Support}(h_j)} s_i\right)h_j|\Omega\rangle$$

$$= 0, \tag{35}$$

where we used the fact that $h_j$ automatically commutes with $s_i$ outside of its support. This indicates that $\left(\prod_i s_i\right)|\Omega\rangle$ is an eigenstate of the new local terms $S(j)^{-1} h_j S(j)^{-1}$ with eigenvalue 0. Furthermore, $S(j)$ is positive and so $S(j)^{-1}$ is also positive. Together with $h_j$ being

Hermitian and having zero as its smallest eigenvalue, this implies that $S(j)^{-1}h_j S(j)^{-1}$ also has zero as its smallest eigenvalue. Therefore $\prod_i s_i |\Omega\rangle$ is a ground state of the new Hamiltonian.

As an example, consider how the deformed toric code [28] would arise from this formalism. We start with the regular toric code, but we shift the energy terms so that their minimum eigenvalues are zero:

$$H = \sum_v (1 - A_v) + \sum_p (1 - B_p),$$

for $A_v = \prod_{i \in \mathrm{star}(v)} \sigma_i^x$ and $B_p = \prod_{i \in p} \sigma_i^z$. Then we take $s_i = e^{\beta \sigma_i^z/2}$, so that the (unnormalized) ground states are $S(\beta)|GS(0)\rangle = \left( \prod_i e^{\beta \sigma_i^z/2} \right)|GS(0)\rangle$. The deformed Hamiltonian is then given by

$$\tilde{H}(\beta) = \sum_v \Big( \prod_{i \in \mathrm{star}(v)} e^{-\beta \sigma_i^z/2} \Big)(1 - \prod_{j \in \mathrm{star}(v)} \sigma_i^x)\Big( \prod_{i \in \mathrm{star}(v)} e^{-\beta \sigma_i^z/2} \Big)$$
$$+ \sum_p \Big( \prod_{i \in p} e^{-\beta \sigma_i^z/2} \Big)(1 - B_p)\Big( \prod_{i \in p} e^{-\beta \sigma_i^z/2} \Big).$$

Then $e^{-\beta \sigma_i^z/2}$ commutes with every term except $\sigma_i^x$, for which $\sigma_i^z \to -\sigma_i^z$ under commutation, giving

$$\tilde{H}(\beta) = \sum_v (e^{-\beta \sum_{i \in \mathrm{star}(v)} \sigma_i^z} - \prod_{j \in \mathrm{star}(v)} \sigma_i^x) + \sum_p \Big( \prod_{i \in p} e^{-\beta \sigma_i^z} \Big)(1 - B_p).$$

This expression differs from the deformed toric code Hamiltonian given in Equation (3) only in that there is a factor of $\prod_{i \in p} e^{-\beta \sigma_i^z}$ at the front of the plaquette term (along with a constant shift). However, this does not change the ground states, which satisfy $B_p = 1$.

A significant question is whether the deformation described by Equation 33 is always enough to ensure the presence of deformed ribbon operators, similar to those that we found for the deformed toric code. Suppose that the undeformed commuting projector model has a ribbon operator $R(t)$ which is topological and only excites the energy terms at the end-points of $t$. Then we define a deformed ribbon operator by

$$\tilde{R}(t) = \Big( \prod_{j \in \mathrm{Support}(R(t))} s_j \Big) R(t) \Big( \prod_{j \in \mathrm{Support}(R(t))} s_j^{-1} \Big). \tag{36}$$

Because the operators $s_j$ commute with $R(t)$ for $j$ outside the support of $R(t)$ and so the factors of $s_j$ and $s_j^{-1}$ would cancel, we can extend the products over $j$ in the support of $R(t)$ to a product over all degrees of freedom, giving us

$$\tilde{R}(t) = \Big( \prod_j s_j \Big) R(t) \Big( \prod_j s_j^{-1} \Big). \tag{37}$$

Then we apply this deformed operator on a deformed ground state $\left( \prod_i s_i \right)|\Omega\rangle$, or on the unexcited region of a general state, so we have

$$\tilde{R}(t)\Big( \prod_i s_i \Big)|\Omega\rangle = \Big( \prod_j s_j \Big) R(t) \Big( \prod_j s_j^{-1} \Big)\Big( \prod_i s_i \Big)|\Omega\rangle$$
$$= \Big( \prod_j s_j \Big) R(t) |\Omega\rangle.$$

Then because $R(t)$ is topological on the undeformed state $|\Omega\rangle$ (as long as we do not deform it over any excitations) we can deform $t$ to $t'$ to obtain

$$\tilde{R}(t)\Big( \prod_i s_i \Big)|\Omega\rangle = \Big( \prod_j s_j \Big) R(t') |\Omega\rangle,$$

and reversing all the steps we get

$$\tilde{R}(t)\big(\prod_i s_i\big)|\Omega\rangle = \tilde{R}(t')\big(\prod_i s_i\big)|\Omega\rangle \,,$$

That is, the deformed ribbon operator is topological when acting on the deformed ground state. This automatically means that it cannot excite any energy terms away from the endpoints of $t$, because we can deform the ribbon away from local energy terms without affecting its action. We can also show this more directly by applying energy terms to the state

$$\tilde{R}(t)\big(\prod_i s_i\big)|\Omega\rangle = \big(\prod_i s_i\big)R(t)|\Omega\rangle \,.$$

For an energy term $\tilde{h}_j = S(j)^{-1}h_j S(j)^{-1}$, we have

$$\tilde{h}_j\tilde{R}(t)\big(\prod_i s_i\big)|\Omega\rangle = S(j)^{-1}h_j S(j)^{-1}\big(\prod_i s_i\big)R(t)|\Omega\rangle$$
$$= S(j)^{-1}\big(\prod_{i\notin\text{Support}(h_j)} s_i\big)h_j R(t)|\Omega\rangle \,.$$

For undeformed energy terms $h_j$ which commute with the original ribbon operator $R(t)$, we have $h_j R(t)|\Omega\rangle = R(t)h_j|\Omega\rangle = 0$, meaning that $\tilde{h}_j\tilde{R}(t)\big(\prod_i s_i\big)|\Omega\rangle = 0$ and so the energy term $\tilde{h}_j$ is unexcited. In other words, the deformed ribbon operator can only excite the deformed energy terms that correspond to the terms that the undeformed ribbon operator excites in the undeformed model.

## 5.1 Quantum double models

One interesting class of commuting projector Hamiltonians that we can study using this formalism is Kitaev's quantum double model [9]. The quantum double model is a generalisation of the toric code, where the directed edges of a lattice are labeled by the elements of a general discrete group, rather than $\mathbb{Z}_2$ as for the toric code

The Hamiltonian is then a sum of commuting projector terms, with one for each vertex and plaquette:

$$H = -\sum_v A_v - \sum_p B_p \,. \tag{38}$$

The plaquette term enforces flatness on the plaquettes: $B_p$ is one when the path element around the plaquette is $1_G$ and zero when the path element is non-trivial. That is

$$B_p = \delta(\hat{g}_p, 1_G), \tag{39}$$

where $\hat{g}_p$ is the path label of the path around the plaquette and $\delta$ is the Kronecker delta.

The vertex term is a sum of gauge transforms:

$$A_v = \frac{1}{|G|}\sum_{g\in G} A_v^g \,, \tag{40}$$

where $A_v^g$ acts on the edges $i$ attached to $v$ according to

$$A_v^g : g_i = \begin{cases} g\,g_i\,, & \text{if } i \text{ points away from } v, \\ g_i g^{-1}, & \text{if } i \text{ points towards } v. \end{cases} \tag{41}$$

For an Abelian group, there are $|G|^2$ ground states on the torus. These are in one-to-one correspondence with the topological charges, or types of excitations. There are $|G|$ pure electric

excitations, one for each irrep of $G$. There are also $|G|$ pure magnetic excitations, one for each element of $G$. These magnetic and electric charges are independent, so a general dyonic excitation can have any value of charge and magnetic flux, leading to $|G|^2$ types of anyon. For a non-Abelian group the picture is slightly more complicated. Firstly, the magnetic flux is only conserved within a conjugacy class. Secondly, the electric charge an excitation can carry depends on its magnetic flux: instead of carrying irreps of the full group, the excitations carry irreps of the centralizer of the magnetic flux label. This leads to fewer types of topological charge, but they have internal spaces allowing for non-Abelian statistics.

Having briefly described the quantum double model [9], we next consider how we should deform it in order to have a phase transition while maintaining the ground state degeneracy. For the toric code, Ref. [28] takes the linear $\sigma_i^z$ term that would cause the condensation phase transition and exponentiates it to obtain the filtering term $S(\beta)$. During the phase transition, all of the electric excitations are condensed and the magnetic ones are confined. For the quantum double model, there are more possibilities because we can condense a subset of the electric excitations or magnetic ones. The ways to do so are described in Ref. [40]. For example, to condense the magnetic excitations in a subgroup $H$, we would add a term $-\frac{\alpha}{|H|}\sum_{h\in H}L_i^h$, where $L_i^h$ multiplies the label of edge $i$ by $h$ and $\alpha$ is a coupling coefficient. We see that the $L_i^h$ term generates non-trivial fluxes (and also confines electric excitations labelled by irreps that are non-trivial in this subgroup).

Instead of condensing the magnetic excitations, in order to compare to our work on the toric code, it will be more convenient to consider condensing electric excitations (and so confining magnetic ones). To do so Ref. [40] introduces a subgroup $M$ and a term

$$T_i^M = \delta(\hat{g}_i \in M) = \sum_{m\in M}\delta(\hat{g}_i, m), \tag{42}$$

which is added to the Hamiltonian with a negative coefficient. This term is clearly Hermitian and a projector. For simplicity, we will take $M$ to be a normal subgroup (these are Case I systems in Ref. [40]). In this case there is a simple interpretation of the term $T_i^M$ as a string tension, which has the effect of confining magnetic excitations with label outside of $M$ and condensing electric excitations labelled by irreps that are trivial in $M$ when it becomes large enough [40]. For example, if we take $M$ to be the trivial subgroup $\{1_G\}$ we can see that the term promotes the trivial state where all of the edges are labeled by $1_G$, condensing all of the electric excitations and confining all of the magnetic ones. For the $\mathbb{Z}_2$ case, which corresponds to the toric code, and denoting the identity element by $+1$, we see that $T_i^M = \delta(\hat{g}_i, +1) = (\sigma_i^z + 1)/2$. This matches the familiar linear field term up to a constant shift and rescaling.

Now instead of introducing $T_i^M$ as a linear term, we use it as a filtering term as described at the start of Section 5. That is, we introduce the Hamiltonian

$$\tilde{H}(\beta) = \sum_j S(j)^{-1}h_j S(j)^{-1}, \tag{43}$$

where $S(j) = \prod_{i\in\text{Support}(h_j)}s_i(\beta)$ for

$$s_i(\beta) = e^{\beta T_i^M/2}. \tag{44}$$

This Hamiltonian will have ground states $S(\beta)|GS(0)\rangle$, where $|GS(0)\rangle$ is a ground state of the undeformed Hamiltonian and

$$S(\beta) = \prod_i s_i(\beta).$$

The energy terms $h_j$ of the original Hamiltonian are $1 - A_v$ and $1 - B_p$, where we have shifted the terms to ensure that 0 is their lowest eigenvalue. This gives us

$$\tilde{H}(\beta) = \sum_v \Big( \prod_{i \in \text{star}(v)} e^{-\beta T_i^M/2} \Big)(1 - A_v)\Big( \prod_{i \in \text{star}(v)} e^{-\beta T_i^M/2} \Big)$$
$$+ \sum_p \Big( \prod_{i \in p} e^{-\beta T_i^M/2} \Big)(1 - B_p)\Big( \prod_{i \in p} e^{-\beta T_i^M/2} \Big).$$

The plaquette terms are diagonal in the group element (configuration) basis, as are the $T_i^M$. This allows us to commute the exponential terms together to get

$$\tilde{H}(\beta) = \sum_v \Big( \prod_{i \in \text{star}(v)} e^{-\beta T_i^M/2} \Big)(1 - A_v)\Big( \prod_{i \in \text{star}(v)} e^{-\beta T_i^M/2} \Big) + \sum_p \Big( \prod_{i \in p} e^{-\beta T_i^M} \Big)(1 - B_p).$$

Now, just as we discussed for the toric code, the exponential term $\prod_{i \in p} e^{-\beta T_i^M}$ in front of $(1 - B_p)$ does not affect the ground states with $B_p = 1$, so we can remove that exponential term and the constant shift to obtain a different Hamiltonian with the same ground states (or at least one that shares the ground states of the form $S(\beta)|GS(0)\rangle$),

$$H(\beta) = \sum_v \Big( \prod_{i \in \text{star}(v)} e^{-\beta T_i^M/2} \Big)(1 - A_v)\Big( \prod_{i \in \text{star}(v)} e^{-\beta T_i^M/2} \Big) - \sum_p B_p.$$

Unlike for the toric code case, bringing the second factor of $e^{-\beta T_i^M/2}$ to the front of the vertex term does not greatly simplify the term, so we will leave the vertex terms in this form. We will mostly be interested in the ground states rather than the Hamiltonian anyway. By the general reasoning from Section 5, the states

$$S(\beta)|GS(0)\rangle = \Big( \prod_i s_i(\beta) \Big)|GS(0)\rangle \,, \tag{45}$$

are ground states of this Hamiltonian, where $|GS(0)\rangle$ is a ground state of the undeformed quantum double model.

One significant feature we found for the deformed toric code is that the 't Hooft loop expectation value obeys a perimeter law even beyond the phase transition to the trivial phase (as we prove in Appendix C). As we show here, a similar result holds for these deformed quantum double models, with the closed undeformed magnetic ribbon operators corresponding to fluxes outside the subgroup $M$ (which we expect to be confined) obeying a perimeter law while those corresponding to fluxes in $M$ (which we expect to be unconfined) obey a zero law. To see this, consider the expectation value of a contractible closed magnetic ribbon operator $C^h(\sigma)$ in one of these ground states. Here $\sigma$ is a ribbon, which is described by a direct path and a dual path (an example of such a closed ribbon is shown in Figure 4). The direct path starts at some vertex, which we call $s.p(\sigma)$, and passes along the edges of the lattice. The dual path starts at a plaquette adjacent to $s.p(\sigma)$ and passes along the dual lattice, cutting through the edges of the lattice. The magnetic ribbon operator acts on the edges along the dual path according to [9]

$$C^h(\sigma) : \hat{g}_i = \begin{cases} g(s.p(\sigma) - v_i)^{-1} h g(s.p(\sigma) - v_i)\hat{g}_i, & \text{if } i \text{ points away from direct path}, \\ \hat{g}_i g(s.p(\sigma) - v_i)^{-1} h^{-1} g(s.p(\sigma) - v_i), & \text{if } i \text{ points towards direct path}. \end{cases}$$

Then the expectation value for the closed ribbon operator is

$$\langle C^h(\sigma) \rangle = \frac{\langle GS(0)| S(\beta)C^h(\sigma)S(\beta)|GS(0)\rangle}{\langle GS(0)| S(\beta)^2 |GS(0)\rangle}$$
$$= \frac{\langle GS(0)| (\prod_i e^{\beta T_i^M/2})C^h(\sigma)(\prod_i e^{\beta T_i^M/2})|GS(0)\rangle}{\langle GS(0)| S(\beta)^2 |GS(0)\rangle}.$$

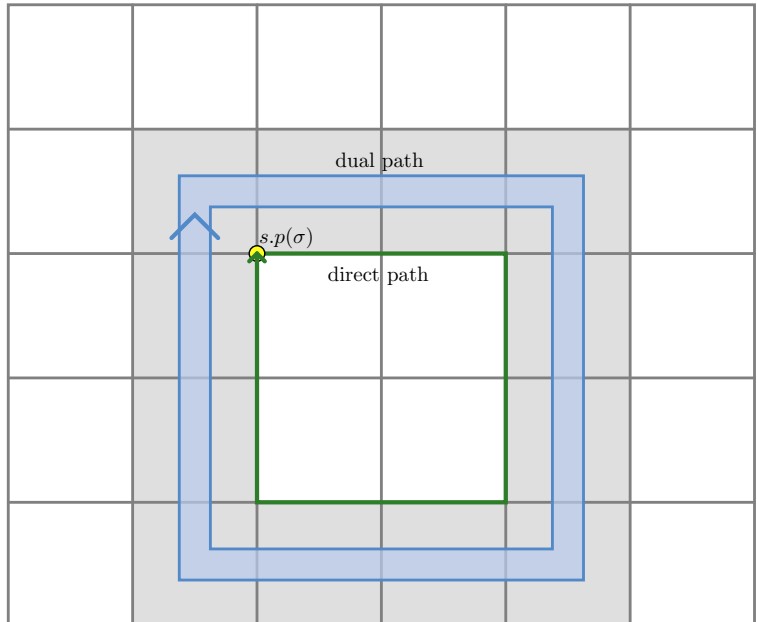

Figure 4: A simple example of a closed ribbon. The yellow vertex is the start-point, $s.p(\sigma)$, which is at the start of the direct path (the green path), while the blue arrow is the dual path. The magnetic ribbon operator alters the edge label of the edges cut by the dual path.

In order to evaluate (or put limits on) this expression, we need to know the commutation relations between $C^h(\sigma)$ and $T_i^M$. We have

$$T_i^M C^h(\sigma) = \delta(\hat{g}_i \in M) C^h(\sigma)$$
$$= C^h(\sigma)\delta((C^h(\sigma): \hat{g}_i) \in M)$$
$$= \begin{cases} C^h(\sigma)\delta(g(s.p(\sigma) - v_i)^{-1} h g(s.p(\sigma) - v_i)\hat{g}_i \in M), & \text{if } i \in \sigma \text{ and points} \\ & \quad \text{away from direct path,} \\ C^h(\sigma)\delta(\hat{g}_i g(s.p(\sigma) - v_i)^{-1} h^{-1} g(s.p(\sigma) - v_i) \in M), & \text{if } i \in \sigma \text{ and points} \\ & \quad \text{towards direct path,} \\ C^h(\sigma)\delta(\hat{g}_i \in M), & \text{otherwise.} \end{cases}$$

This means that $C^h(\sigma)$ commutes with $T_i^M$ for $i \notin \sigma$. Now, we separate two different cases, one where $h$ is in the subgroup $M$ and one where it is not. If $h \in M$, then so is $g(s.p(\sigma) - v_i)^{-1} h^{\pm 1} g(s.p(\sigma) - v_i)$ because $M$ is normal. This means that

$$\delta(g(s.p(\sigma) - v_i)^{-1} h g(s.p(\sigma) - v_i)\hat{g}_i \in M) = \delta(m'\hat{g}_i \in M),$$

for some $m' \in M$ and so

$$\delta(g(s.p(\sigma) - v_i)^{-1} h g(s.p(\sigma) - v_i)\hat{g}_i \in M) = \delta(\hat{g}_i \in M),$$

from the subgroup property. A similar result holds for

$$\delta(\hat{g}_i g(s.p(\sigma) - v_i)^{-1} h^{-1} g(s.p(\sigma) - v_i) \in M).$$

Therefore $C^h(\sigma)$ commutes with all $T_i^M$ for $h \in M$ and so commutes with all of the $e^{\beta T_i^M/2}$ in $S(\beta)$. This means that, for an element $m \in M$, we have

$$\langle C^m(\sigma) \rangle = \frac{\langle GS(0)| S(\beta) C^m(\sigma) S(\beta) |GS(0) \rangle}{\langle GS(0)| S(\beta)^2 |GS(0) \rangle} = \frac{\langle GS(0)| S(\beta)^2 C^m(\sigma) |GS(0) \rangle}{\langle GS(0)| S(\beta)^2 |GS(0) \rangle}.$$

Then $C^m(\sigma)$ acts trivially on the ground state for a contractible closed ribbon $\sigma$:

$$C^m(\sigma)|GS(0)\rangle = |GS(0)\rangle \,.$$

Therefore,

$$\langle C^m(\sigma) \rangle = \frac{\langle GS(0)|S(\beta)^2|GS(0)\rangle}{\langle GS(0)|S(\beta)^2|GS(0)\rangle} = 1 \,.$$

In other words, the 't Hooft loops for fluxes in the group $M$ satisfy a zero law, which matches our expectation that these fluxes are unconfined from the linear term case considered in Ref. [40].

Now consider the case where the flux label $h$ is outside of $M$. In this case $g(s.p(\sigma)-v_i)^{-1}hg(s.p(\sigma)-v_i)$ is also outside of $M$ and so

$$\delta(g(s.p(\sigma)-v_i)^{-1}hg(s.p(\sigma)-v_i)\hat{g}_i \in M) = \delta(\hat{g}_i \in q_{h,i}M)\,,$$

for some non-trivial coset $q_{h,i}M$. We therefore have

$$T_i^M C^h(\sigma) = \begin{cases} C^h(\sigma)\delta(\hat{g}_i \in q_{h,i}M)\,, & \text{if } i \in \sigma,\\ C^h(\sigma)\delta(\hat{g}_i \in M)\,, & \text{otherwise,} \end{cases}$$

where $q_{h,i}$ can depend on the orientation of $i$ as well as the path from the start-point of $\sigma$ to the edge. This gives us an expectation value for the 't Hooft loop of

$$
\begin{aligned}
\langle C^h(\sigma) \rangle &= \frac{\langle GS(0)|(\prod_i e^{\beta T_i^m/2})C^h(\sigma)(\prod_i e^{\beta T_i^M/2})|GS(0)\rangle}{\langle GS(0)|S(\beta)^2|GS(0)\rangle}\\
&= \frac{\langle GS(0)|(\prod_{i\notin\sigma} e^{\beta T_i^m})(\prod_{i\in\sigma} e^{\beta T_i^m/2})C^h(\sigma)(\prod_{i\in\sigma} e^{\beta T_i^M/2})|GS(0)\rangle}{\langle GS(0)|S(\beta)^2|GS(0)\rangle}\\
&= \frac{\langle GS(0)|(\prod_i e^{\beta T_i^m})(\prod_{i\in\sigma} e^{-\beta T_i^m/2})C^h(\sigma)(\prod_{i\in\sigma} e^{\beta T_i^M/2})|GS(0)\rangle}{\langle GS(0)|S(\beta)^2|GS(0)\rangle}\\
&= \frac{\langle GS(0)|(\prod_i e^{\beta T_i^m})(\prod_{i\in\sigma} e^{-\beta\delta(\hat{g}_i\in M)/2})C^h(\sigma)(\prod_{i\in\sigma} e^{\beta\delta(\hat{g}_i\in M)/2})|GS(0)\rangle}{\langle GS(0)|S(\beta)^2|GS(0)\rangle}\\
&= \frac{\langle GS(0)|(\prod_i e^{\beta T_i^m})(\prod_{i\in\sigma} e^{-\beta\delta(\hat{g}_i\in M)/2})(\prod_{i\in\sigma} e^{\beta\delta(\hat{g}_i\in q_{h,i}^{-1}M)/2})C^h(\sigma)|GS(0)\rangle}{\langle GS(0)|S(\beta)^2|GS(0)\rangle} \,.
\end{aligned}
$$

Then $C^h(\sigma)|GS(0)\rangle = |GS(0)\rangle$ from the properties of the undeformed model and so

$$
\begin{aligned}
\langle C^h(\sigma) \rangle &= \frac{\langle GS(0)|(\prod_i e^{\beta T_i^m})(\prod_{i\in\sigma} e^{-\beta\delta(\hat{g}_i\in M)/2})(\prod_{i\in\sigma} e^{\beta\delta(\hat{g}_i\in q_{h,i}^{-1}M)/2})|GS(0)\rangle}{\langle GS(0)|S(\beta)^2|GS(0)\rangle}\\
&= \frac{\langle GS(0)|S(\beta)^2(\prod_{i\in\sigma} e^{-\beta\delta(\hat{g}_i\in M)/2+\beta\delta(\hat{g}_i\in q_{h,i}^{-1}M)/2})|GS(0)\rangle}{\langle GS(0)|S(\beta)^2|GS(0)\rangle} \,.
\end{aligned}
$$

We can find a simple limit for this expression by expanding the undeformed ground state in terms of the group element basis:

$$|GS(0)\rangle = \sum_{\{g_i\}} a_{\{g_i\}} |\{g_i\}\rangle \,,$$

where the value of the coefficients $a_{\{g_i\}}$ will not matter for this discussion. Then the numerator in the expression for the expectation value is given by

$$
\begin{aligned}
&\langle GS(0)|S(\beta)^2\left(\prod_{i\in\sigma} e^{-\beta\delta(\hat{g}_i\in M)/2+\beta\delta(\hat{g}_i\in q_{h,i}^{-1}M)/2}\right)|GS(0)\rangle\\
&= \sum_{\{g_i'\},\{g_i\}} a^*_{\{g_i'\}}a_{\{g_i\}} \langle\{g_i'\}|\left(\prod_i e^{\sum_i \beta\delta(g_i\in M)}\right)\left(\prod_{i\in\sigma} e^{-\beta\delta(g_i\in M)/2+\beta\delta(g_i\in q_{h,i}^{-1}M)/2}\right)|\{g_i\}\rangle \,,
\end{aligned}
$$

where we have used the fact that all the operators in this expression are diagonal in the group element basis to replace all the operators with their eigenvalues. Then we can use the orthogonality of the basis vectors to obtain

$$\langle GS(0)|S(\beta)^2 \left( \prod_{i\in\sigma} e^{-\beta\delta(\hat{g}_i\in M)/2+\beta\delta(\hat{g}_i\in q_{h,i}^{-1}M)/2} \right)|GS(0)\rangle$$
$$= \sum_{\{g_i\}} |a_{\{g_i\}}|^2 \left( \prod_i e^{\sum_i \beta\delta(g_i\in M)} \right) \left( \prod_{i\in\sigma} e^{-\beta\delta(g_i\in M)/2+\beta\delta(g_i\in q_{h,i}^{-1}M)/2} \right).$$

Next, note that each term in the sum is non-negative (because it is a product of exponentials, which are positive for real arguments, with $|a_{\{g_i\}}|^2$). In addition,

$$e^{-\beta\delta(g_i\in M)/2+\beta\delta(g_i\in q_{h,i}^{-1}M)/2}$$

is bounded from below by $e^{-|\beta|/2}$. Therefore,

$$\sum_{\{g_i\}} |a_{\{g_i\}}|^2 \left( \prod_i e^{\sum_i \beta\delta(g_i\in M)} \right) \left( \prod_{i\in\sigma} e^{-\beta\delta(g_i\in M)/2+\beta\delta(g_i\in q_{h,i}^{-1}M)/2} \right)$$
$$\geq \sum_{\{g_i\}} |a_{\{g_i\}}|^2 \left( \prod_i e^{\sum_i \beta\delta(g_i\in M)} \right) e^{-|\beta|L_\sigma/2},$$

where $L_\sigma$ is the length of the ribbon $\sigma$. We can also write the denominator in the expression for the expectation value as

$$\sum_{\{g_i\}} |a_{\{g_i\}}|^2 \left( \prod_i e^{\sum_i \beta\delta(g_i\in M)} \right).$$

Therefore, the entire expectation value satisfies the inequality

$$\langle C^h(\sigma)\rangle \geq \frac{e^{-|\beta|L_\sigma/2} \sum_{\{g_i\}} |a_{\{g_i\}}|^2 \left( \prod_i e^{\sum_i \beta\delta(g_i\in M)} \right)}{\sum_{\{g_i\}} |a_{\{g_i\}}|^2 \left( \prod_i e^{\sum_i \beta\delta(g_i\in M)} \right)}$$
$$\geq e^{-|\beta|L_\sigma/2}. \tag{46}$$

This indicates that the expectation value can decay at most as quickly as the length of the closed ribbon and so the 't Hooft loop satisfies a perimeter law, just like for the deformed toric code.

## 6 Conclusion

In this study, we have investigated the behavior of anyonic excitations in a deformed toric code model that allows for a transition from a topological phase to a trivial phase. Previous studies of this model focused solely on ground states and neglected anyonic excitations. In addition, to better understand the relationship between 1-form symmetry and topological order in this model, we examined the behaviors of the Wilson loop and 't Hooft loop operators across the transition.

Our analysis revealed that magnetic excitations in both the topological and trivial phases can be described using a non-unitary deformed magnetic ribbon operator. We found that the electric charges condense across the transition as we may expect, but the accompanying confinement of the magnetic charges is less obvious. There exist non-unitary magnetic ribbon

operators that move magnetic charges without any energy cost in both phases. However, applying a semi-infinite non-unitary ribbon operator to separate magnetic charges in the trivial phase results in a wave function with a zero norm in the thermodynamic limit, which provides an alternative mechanism for confinement. This indicates that the condensation and confinement of the topological charges still provides a useful way to characterise the transition to the trivial phase. Our study provides a deeper understanding of the behavior of anyonic excitations in the deformed toric code model and demonstrates that we must consider this alternate confinement mechanism of magnetic charges in the trivial phase.

Another way in which we examined the condensation and confinement is through the Wilson and 't Hooft loops, which act as topological charge measurement operators. We found that the Wilson loop obeys a zero law on both sides of the transition, meaning that its expectation value in the ground state does not decay with the size of the loop. On the other hand, the 't Hooft loop obeys a perimeter law on both sides, with its expectation value decaying exponentially with the length of the loop. While this suggests that electric charge is present in the ground state (implying some degree of condensation), we might expect an area law in the trivial phase (which we would obtain if we applied a linear magnetic field [41]). This implies that these loop operators are not good descriptors for the phase transition or the condensation in this case. As we showed in Section 5.1, this holds not just for the deformed toric code model, but also for other deformed quantum double models.

Another feature of the model is that there are four degenerate ground states on the torus in both sides of the transition. While this was known previously, we have interpreted this as a pattern of 1-form symmetry breaking. This appears to be in contradiction to the idea that the 1-form symmetry breaking immediately corresponds to a topological order. We have argued that, in order to have a topological order (indistinguishable degenerate ground states), one needs the presence of an additional 1-form symmetry that acts as an order parameter for the other 1-form symmetry. This can be thought of as a mixed 't Hooft anomaly that is directly related to the non-trivial braiding of anyons. In the deformed toric model, the topological phase ($\beta < \beta_c$) has an emergent mixed 't Hooft anomaly akin to the original toric code model, where the deformed 't Hooft loop acts both as an emergent 1-form symmetry and an order parameter for the Wilson loop. On the other hand, the trivial phase ($\beta > \beta_c$) loses such an anomaly as the deformed 't Hooft loop ceases to be a 1-form symmetry, even though it still leads to spontaneous symmetry breaking of the Wilson loop 1-form symmetry. Hence the spontaneous symmetry breaking of the Wilson loop 1-form symmetry alone does not necessarily lead to topological order. Instead, the presence of a mixed 't Hooft anomaly associated with both 1-form symmetries (Wilson and 't Hooft) is crucial for the emergence of topological order. We speculate that this would be the case for all higher-form symmetries in gapped systems.

The findings of this study raise intriguing questions about the relationship between higher-form symmetry breaking and topological phases. Future research could explore this relationship in greater detail, particularly in more generic models, given that the model studied here is finely tuned. Specifically, it is still being determined whether the type of confinement or spontaneous symmetry breaking observed in this study would be present in more generic models.

In Section 5, we pointed out that the deformed toric code can be generalized to many different commuting projector models, as well as different types of deformation. One way to extend the deformed toric code model is to allow for inhomogeneous deformations (see Appendix D), meaning that different parts of the lattice could have parameters on either side of the phase transition. In this case, the ground states can still be found exactly. Studying this model further could improve our understanding of the phase transition and boundary modes. In addition, it may be possible to change the spatial dependence of the parameter with time in order to control the behaviour of the excitations.

Another promising avenue for future research is to examine a similar model based on the X-cube model. Such "deformed X-cube" models have already been constructed from both the Hamiltonian [44] and tensor network [45] perspectives and can be tuned between four phases: stacks of decoupled toric codes, a 3+1d toric code, the X-cube model and the trivial paramagnet. It would be interesting to study the fate of the generalized symmetries during these deformations. These future investigations could help shed light on the behavior of topological phases and higher-form symmetries in more complex and varied models.

## Acknowledgments

We are grateful to S. H. Simon and P. Fendley for fruitful discussions about the deformed toric code model. We also thank C. Castelnovo and S. Pace for comments on our manuscript. This work was supported by the Natural Sciences and Engineering Research Council (NSERC) of Canada and the Center for Quantum Materials at the University of Toronto (J.H. and Y.B.K.).

**Funding information**   Our collaboration is a part of the effort in the Advanced Study Group on "Entanglement and Dynamics in Quantum Matter" in the Center for Theoretical Physics of Complex Systems at the Institute for Basic Science (D.X.N. and Y.B.K.). Y.B.K. is further supported by the Simons Fellowship from the Simons Foundation and the Guggenheim Fellowship from the John Simon Guggenheim Memorial Foundation. D.X.N. is supported by Grant No. IBS-R024-D1.

## A   Tensor network description

As we described in Section 3.1, the confinement present in the deformed toric code model appears similar in character to the confinement described in Ref. [29] in the tensor network formalism. In fact, Ref. [29] uses tensor networks to study filtered toric code states, which match the (even-even) deformed toric code ground state for a particular family of parameters. Here we will reproduce the tensor network representation of the deformed toric code ground state and demonstrate that the Matrix Product Operators (MPOs) in the tensor network, which produce anyons, are equivalent to the non-unitary deformed magnetic ribbon operators in the Hamiltonian description.

In a tensor network, a physical state is represented by a layer of tensors. These tensors have two types of indices: physical and virtual. The virtual indices lie in the layer of tensors and facilitate the matrix multiplication of the tensors when summed over (we can think of these as edges or legs connecting the tensors). Contracting these indices gives a quantity which depends only on the physical indices, and represents the amplitude for that configuration of physical indices in the overall state. For example, the physical indices could label a configuration of spins, and the amplitude for that configuration of spins in the overall state is obtained by matrix multiplication of the tensors, contracting over the virtual indices. For a 1d tensor network, for instance, the amplitude for a configuration $|\{i\}\rangle$ in a state described by tensors $T^{i_n}_{\alpha_n,\alpha_{n+1}}$ would be

$$\langle \{i\}|\psi\rangle = \sum_{\{\alpha_n\}} \prod_{n=1}^{N} T^{i_n}_{\alpha_n,\alpha_{n+1}}. \tag{A.1}$$

While the space occupied by the physical indices is fixed by the local degrees of freedom in the physical system, the virtual indices exist in a different space. The number of values which these indices can take is called the bond dimension. Often a tensor network state is

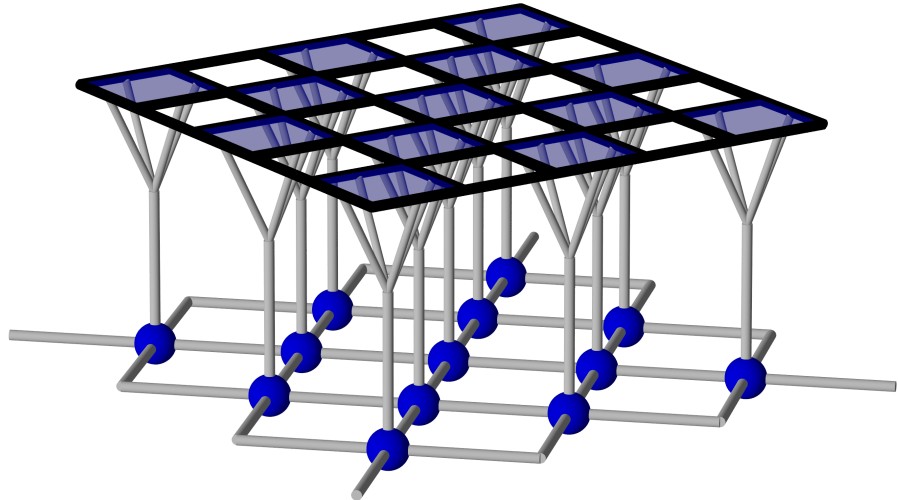

Figure 5: A visual representation of the toric code tensor network. The upper layer is the physical layer, consisting of the spins of the toric code, while the lower layer includes the virtual degrees of freedom which are contracted to obtain the ground state. The spins on the boundaries of alternating plaquettes (shaded) in the physical layer are grouped and connected to the tensors (blue spheres).

only an approximation to a desired ground state and increasing the bond dimension improves this approximation. However, in the case of the deformed toric code, the ground state can be represented exactly by a tensor network and the bond dimension required is only two.

In order to construct the tensor network for the deformed toric code, we first consider the representation of the ordinary toric code (see, e.g., Ref. [46]). As shown in Figure 5, the spins on alternating plaquettes are grouped and connected to a tensor, with the neighbouring tensors connected by virtual legs in a square lattice. This means that each tensor has four physical indices, corresponding to the four spins on the plaquette, and four virtual indices connecting them to the tensors representing diagonally adjacent plaquettes. The tensor is then given by

$$A_{\alpha_1,\alpha_2,\alpha_3,\alpha_4}^{i_1,i_2,i_3,i_4} = \delta(i_1, \alpha_2\alpha_1^{-1})\delta(i_2, \alpha_3\alpha_2^{-1})\delta(i_3, \alpha_4\alpha_3^{-1})\delta(i_4, \alpha_1\alpha_4^{-1}). \tag{A.2}$$

Here the $i$ variables are physical indices and the $\alpha$ variables are virtual indices. Both types take the values $\pm 1$, with $\mathbb{Z}_2$ group multiplication. We can think of the physical indices as connecting two adjacent virtual indices: in order to go from one virtual index to the next, we must multiply it by the appropriate physical label. If this condition is satisfied for all indices, the tensor is one, otherwise it is zero. Because each physical index describes the change in virtual index, the tensor has a symmetry where we can multiply each virtual index by $-1$ without affecting the value of the tensor:

$$
\begin{aligned}
A_{-\alpha_1,-\alpha_2,-\alpha_3,-\alpha_4}^{i_1,i_2,i_3,i_4} \\
&= \delta(i_1,(-\alpha_2)(-\alpha_1)^{-1})\delta(i_2,(-\alpha_3)(-\alpha_2)^{-1})\delta(i_3,(-\alpha_4)(-\alpha_3)^{-1})\delta(i_4,(-\alpha_1)(-\alpha_4)^{-1}) \\
&= \delta(i_1, \alpha_2\alpha_1^{-1})\delta(i_2, \alpha_3\alpha_2^{-1})\delta(i_3, \alpha_4\alpha_3^{-1})\delta(i_4, \alpha_1\alpha_4^{-1}) \\
&= A_{\alpha_1,\alpha_2,\alpha_3,\alpha_4}^{i_1,i_2,i_3,i_4}.
\end{aligned}
\tag{A.3}
$$

This virtual symmetry gives rise to an MPO, which crosses the virtual legs of tensors and modifies the affected tensors by replacing the index for the crossed leg with its negative. That is, if the MPO crosses tensor

$$A_{\alpha_1,\alpha_2,\alpha_3,\alpha_4}^{i_1,i_2,i_3,i_4},$$

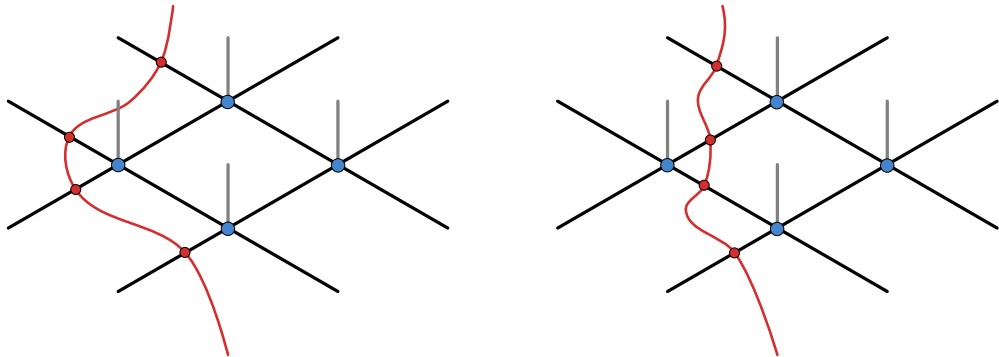

Figure 6: The matrix product operator (red) has the pull-through property, meaning that it can be deformed over a tensor without affecting the state. This means that the left and right diagrams represent the same state.

on the leg labelled by $\alpha_1$, we should replace the tensor with

$$\tilde{A}^{i_1,i_2,i_3,i_4}_{\alpha_1,\alpha_2,\alpha_3,\alpha_4} = A^{i_1,i_2,i_3,i_4}_{-\alpha_1,\alpha_2,\alpha_3,\alpha_4}.$$

Importantly, the MPO should not be regarded as changing the virtual index itself like a regular operator might. Because $\alpha_1$ is a dummy index, changing the variable directly to $-1$ from $+1$ has no effect. Instead the MPO changes the tensor on one side of the leg (not both), giving a different coefficient once $\alpha_1$ is contracted. The virtual symmetry of the tensors gives the MPO a "pull-through" property, meaning that we can deform the position of the MPO across the tensor, as shown in Figure 6. That is, if we apply the MPO on leg 1 we obtain

$$
\begin{aligned}
A^{i_1,i_2,i_3,i_4}_{-\alpha_1,\alpha_2,\alpha_3,\alpha_4} &= A^{i_1,i_2,i_3,i_4}_{-(-\alpha_1),-\alpha_2,-\alpha_3,-\alpha_4} \\
&= A^{i_1,i_2,i_3,i_4}_{\alpha_1,-\alpha_2,-\alpha_3,-\alpha_4},
\end{aligned}
\tag{A.4}
$$

which is equivalent to applying the MPO on legs 2, 3 and 4.

We can see the effect of the MPO on the physical state by examining the modified tensor in more detail. We have

$$
\begin{aligned}
A^{i_1,i_2,i_3,i_4}_{-\alpha_1,\alpha_2,\alpha_3,\alpha_4} &= \delta(i_1, \alpha_2(-\alpha_1)^{-1})\delta(i_2, \alpha_3\alpha_2^{-1})\delta(i_3, \alpha_4\alpha_3^{-1})\delta(i_4, (-\alpha_1)\alpha_4^{-1}) \\
&= \delta(-i_1, \alpha_2\alpha_1^{-1})\delta(i_2, \alpha_3\alpha_2^{-1})\delta(i_3, \alpha_4\alpha_3^{-1})\delta(-i_4, \alpha_1\alpha_4^{-1}) \\
&= A^{-i_1,i_2,i_3,-i_4}_{\alpha_1,\alpha_2,\alpha_3,\alpha_4}.
\end{aligned}
\tag{A.5}
$$

We see that applying the MPO swaps the coefficient for configuration $|i_1, i_2, i_3, i_4\rangle$ with $|-i_1, i_2, i_3, -i_4\rangle$. This is exactly the action of a $\sigma^x$ operator applied on the physical edges 1 and 4. Extending this to a longer MPO, the MPO in the virtual layer corresponds to a magnetic ribbon operator (which is a product of $\sigma^x$ terms along aa ribbon) on the physical layer. The electric ribbon operator, by contrast, is equivalent to locally modifying the tensors at the two ends of the ribbon operator.

Having considered the original toric code, we now examine the deformed toric code. We can construct the appropriate tensor network state by applying the operator $S(\beta) = \prod_{\text{edges } i} e^{\beta\sigma_i^z/2}$, which is a product of single-spin operators, to the toric code tensor network, as shown in Figure 7. This has the effect of modifying the tensors to

$$A^{i_1,i_2,i_3,i_4}_{\alpha_1,\alpha_2,\alpha_3,\alpha_4}(\beta) = e^{\beta(i_1+i_2+i_3+i_4)/2}\delta(i_1, \alpha_2\alpha_1^{-1})\delta(i_2, \alpha_3\alpha_2^{-1})\delta(i_3, \alpha_4\alpha_3^{-1})\delta(i_4, \alpha_1\alpha_4^{-1}), \tag{A.6}$$

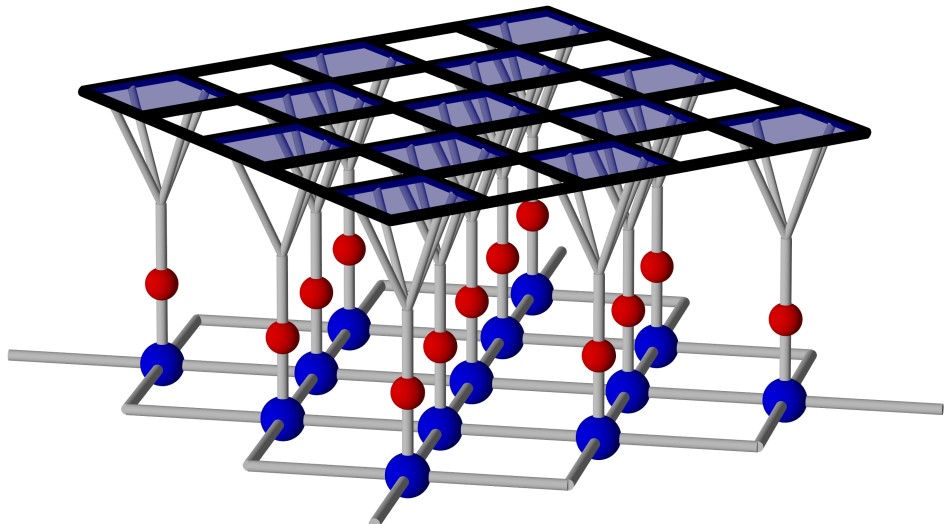

Figure 7: We can obtain the tensor network for the deformed toric code from that of
the regular toric code by applying an additional layer of tensors (smaller red spheres),
which act as single spin operators $e^{\beta \sigma_i^z / 2}$ on the physical degrees of freedom.

which now gives different weights to the different configurations satisfying the Kronecker delta
conditions. This tensor still satisfies the same virtual symmetry

$$A_{\alpha_1, \alpha_2, \alpha_3, \alpha_4}^{i_1, i_2, i_3, i_4}(\beta) = A_{-\alpha_1, -\alpha_2, -\alpha_3, -\alpha_4}^{i_1, i_2, i_3, i_4}(\beta). \tag{A.7}$$

This leads to the network supporting the same MPO as the toric code state: an MPO which
modifies affected tensors by flipping some of the virtual indices. However, the effect of this
MPO on the physical state is not the same. If we apply the MPO on leg 1 of the tensor as
before, we obtain the new tensor

$$
\begin{aligned}
\tilde{A}_{\alpha_1, \alpha_2, \alpha_3, \alpha_4}^{i_1, i_2, i_3, i_4}(\beta) &= A_{-\alpha_1, \alpha_2, \alpha_3, \alpha_4}^{i_1, i_2, i_3, i_4}(\beta) \\
&= e^{\beta(i_1 + i_2 + i_3 + i_4)/2} \delta(-i_1, \alpha_2 \alpha_1^{-1}) \delta(i_2, \alpha_3 \alpha_2^{-1}) \delta(i_3, \alpha_4 \alpha_3^{-1}) \delta(-i_4, \alpha_1 \alpha_4^{-1}) \\
&= e^{\beta(i_1 + i_4)} e^{\beta(-i_1 + i_2 + i_3 - i_4)/2} \delta(-i_1, \alpha_2 \alpha_1^{-1}) \delta(i_2, \alpha_3 \alpha_2^{-1}) \delta(i_3, \alpha_4 \alpha_3^{-1}) \delta(-i_4, \alpha_1 \alpha_4^{-1}) \\
&= e^{\beta(i_1 + i_4)} A_{\alpha_1, \alpha_2, \alpha_3, \alpha_4}^{-i_1, i_2, i_3, -i_4}(\beta).
\end{aligned} \tag{A.8}
$$

Instead of being equivalent to the application of $\sigma_i^x$ operators, this is equivalent to applying
a series of $e^{\beta \sigma_i^z} \sigma_i^x = e^{\beta \sigma_i^z / 2} \sigma_i^x e^{-\beta \sigma_i^z / 2}$ operators. This is the same as the deformed magnetic
ribbon operator from Equation (11), so we see that the deformed ribbon operator can be in-
terpreted as the MPO arising from the preserved virtual symmetry of the underlying tensor
network. This allows us to relate the non-unitary nature of the ribbon operator to the con-
finement described in Ref. [29]. Above the phase transition, the norm of the state obtained by
applying a semi-infinite MPO into the tensor network is zero, implying confinement [29]. Sim-
ilarly, the deformed ribbon operators are non-unitary and so can reduce the norm of the state.
This effect becomes relevant at $\beta$ above the phase transition, allowing a form of confinement
without energetic cost.

## B  Mapping to Ising model

As described in Ref. [28], there is a relationship between the deformed toric code model and the
2d classical Ising model, such that the expectation values of many operators in the deformed

toric code are equal to expectation values of quantities in the finite temperature Ising model. To see why this is so, consider the ground state wavefunction, which is a sum of closed (dual) loop configurations. For the even-even ground state, which has no non-contractible loops, we can treat the loops as domain walls. To do so, we introduce Ising spin variables (taking values $\pm 1$) at the vertices of the lattice, which we denote by $\theta_v$, such that the toric code variable $\sigma_i^z = \theta_{s(i)}\theta_{t(i)}$, where $s(i)$ and $t(i)$ are the two ends of edge $i$. That is, a down spin ($\sigma_i^z = -1$) separates domains with different Ising spin values. This assignment of Ising spins is consistent, as long as the toric code down spins form closed contractible loops (i.e., there are no magnetic fluxes present and we are in the even-even sector). However, the assignment of Ising spins to a toric code configuration is not unique, as multiplying every Ising spin by $-1$ gives the same toric code configuration. Any physical toric code property is the same for either Ising spin assignment, however, so this does not affect any expectation values that we wish to calculate. This redundancy can be regarded as a global gauge $\mathbb{Z}_2$ symmetry.

We can use this mapping to relate the Hamiltonian itself to a quantum Ising-like model (especially in the low $\beta$ case, where we obtain the Ising model in a transverse field) [28, 47], at least in the subspace where the plaquette terms are satisfied. However, for our purposes it is more useful to instead relate the ground state to the classical Ising partition function at zero field for all $\beta$ [28]. Under this mapping of variables, the deformed toric code ground state, which is given by

$$|GS(\beta)\rangle \propto \sum_{\text{loop configurations, } a} \prod_{\text{edges, } i} e^{\beta \sigma_i^z/2}|a\rangle \, ,$$

becomes

$$|GS(\beta)\rangle = \frac{1}{\sqrt{Z(\beta)}} \sum_{\text{Ising configurations, } \{\theta_v\}} \prod_{\text{edges, } i} e^{\beta \theta_{s(i)}\theta_{t(i)}/2}|\{\theta_v\}\rangle \, , \tag{B.1}$$

where

$$Z(\beta) = \sum_{\{\theta_v\}} \prod_i e^{\beta \theta_{s(i)}\theta_{t(i)}} = \sum_{\{\theta_v\}} e^{\sum_i \beta \theta_{s(i)}\theta_{t(i)}}$$

can be recognised as the Ising model partition function, with $\beta$ playing the role of the Ising coupling divided by the temperature. If we wish to calculate the expectation value of some operator which is a function of $\sigma^z$ operators in the deformed toric code, we can relate the expectation value to one in the Ising model [28]:

$$
\begin{aligned}
\langle f(\{\sigma_i^z\})\rangle &= \langle GS(\beta)| f(\{\sigma_i^z\})|GS(\beta)\rangle \\
&= \frac{2}{Z(\beta)} \sum_{\text{loop configurations, } a} \langle a| \prod_j e^{\beta \sigma_j^z/2} f(\{\sigma_i^z\}) \sum_{\text{loop configurations, } b} \prod_i e^{\beta \sigma_i^z/2}|b\rangle \\
&= \frac{2}{Z(\beta)} \sum_{\text{loop configurations, } a,b} \delta(a,b) \langle a| f(\{\sigma_i^z\}) \prod_i e^{\beta \sigma_i^z}|a\rangle \\
&= \frac{1}{Z(\beta)} \sum_{\text{Ising configurations, } \{\theta_v\}} f(\{\theta_{s(i)}\theta_{t(i)}\}) e^{\sum_i \beta \theta_{s(i)}\theta_{t(i)}} \\
&= \langle f(\{\theta_{s(i)}\theta_{t(i)}\})\rangle_{\text{Ising}} \, , \tag{B.2}
\end{aligned}
$$

where the factor of two before switching to Ising variables is to account for the two-to-one mapping from Ising variables to physical toric code configurations.

In particular, note that the electric ribbon operator $L(t) = \prod_{i \in t} \sigma_i^z$ is mapped to the quantity

$$\prod_{i \in t} \theta_{s(i)}\theta_{t(i)} = \theta_{s(t)}\theta_{e(t)} \, ,$$

where $s(t)$ and $e(t)$ are the vertices at the start and end of the path $t$ respectively, because there are two copies of $\theta_v$ for each intermediate vertex (one from each adjacent edge on the



Figure 8: A convenient reference state for the odd-even sector. The solid lines represent down spins and the dashed lines represent up spins.

path), which cancel. This means that the expectation value of the electric ribbon operator is equal to the expectation value of the two Ising spins at the ends of the path, i.e., to a correlation function. We know that well below the phase transition (high temperature in the Ising model) this correlation function tends to zero at large distances, while well above the phase transition (low temperature in the Ising model) this correlation function tends to a positive value, due to the behaviour of the magnetization [48, 49]. This demonstrates that the electric ribbon operator is absorbed into the ground state at high $\beta$, as we claimed in Section 3.1.

Because the mapping to the Ising model relies on only having closed loop configurations, there is no counterpart to the $\sigma_i^x$ operator in the Ising description. However, it is still possible to use the Ising description for combinations of $\sigma_i^x$ operators that do not produce open strings, such as toric code vertex operations $A_v^- = \prod_{i \in \text{star}(v)} \sigma_i^x$. This vertex operator maps onto an Ising spin flip operation. However, we must be careful when calculating expectation values of such operators, because they do not commute with the weight $\prod_i e^{\beta \sigma_i^z}$.

So far, we have considered the even-even ground state of the toric code and its deformed variant. However, it is possible to use a similar picture to describe the other ground states in terms of the Ising model, but with different boundary conditions. In order to do so, we must first define a reference state, which is just a configuration in the desired topological sector (even-even, odd-even, etc.). Any other configuration in the same sector as the reference state can be obtained by applying toric code vertex transforms. Because these transforms are self-inverse, at each vertex $v$ we either apply a transform $A_v^-$ or apply the identity operator $A_v^+ = I$. Therefore we can label each configuration by the sequence of transforms we need to apply in order to reach it from the reference state [28]. The variable describing whether we apply a transform at each vertex or not then becomes an Ising spin in our new description. If we apply a transform on the vertex to obtain the configuration, the Ising spin is $-1$. If we apply the identity, the Ising spin is $+1$. For an Ising spin $\theta_v$, we then denote the relevant transform by $A_v^{\theta_v}$. One small subtlety to this is that the product of all vertex transforms is equal to the identity, meaning that multiplying all Ising spins by $-1$ is trivial in the toric code space and so there are two Ising configurations that give the same toric code configuration (just as we discussed previously for the even-even case). However, because no physical quantity is different between the two equivalent configurations, we are free to average over these or have some convention for removing one as preferred.

Now consider the odd-even sector, where the parity around the horizontal handle of the torus is odd and the parity around the vertical one is even. A convenient reference state for this sector is shown in Figure 8. The (un-normalised) ground state for the odd-even sector is

given by

$$
\begin{aligned}
|GS_{-+}(\beta)\rangle &= \sum_{\{\theta_v\}} e^{\beta \sum_i \sigma_i^z/2} \prod_v A_v^{\theta_v} |\text{ref.}\rangle \\
&= \sum_{\{\theta_v\}} e^{\beta \sum_i \sigma_i^z(\{\theta_v\})/2} \prod_v A_v^{\theta_v} |\text{ref.}\rangle \,,
\end{aligned}
\tag{B.3}
$$

where $\sigma_i^z(\{\theta_v\})$ is the eigenvalue of $\sigma_i^z$ for the state $|\{\theta_v\}\rangle = \prod_v A_v^{\theta_v}|\text{ref.}\rangle$. We therefore need to determine the relationship between the Ising variables $\{\theta_v\}$ and the toric code spin $\sigma_i^z(\{\theta_v\})$. To do so, we first label each vertex with co-ordinates $(x, y)$. The lattice is periodic, so for horizontal size $N_x$ and vertical size $N_y$ we have $N_x + 1 = 1$ and $N_y + 1 = 1$. We also label the edges with the direction ($\hat{x}$ or $\hat{y}$) and the leftmost (for horizontal edges) or lowermost (for vertical edges) vertex $(x, y)$ attached to that edge. In the reference state shown in Figure 8, which has all $\theta_v = 1$ (or all $\theta_v = -1$) by definition of $\theta_v$, we have

$$
\sigma_{\hat{x},(x,y)}^z(\{+\}) = \begin{cases} -1, & \text{if } x = N_x, \\ +1, & \text{otherwise,} \end{cases}
\tag{B.4}
$$

and

$$
\sigma_{\hat{y},(x,y)}^z(\{+\}) = +1, \quad \text{for all} \quad (x, y).
\tag{B.5}
$$

We can use this to find the toric code spin for an arbitrary set of $\theta_v$. We know that applying a vertex transform on either vertex adjacent to an edge flips the spin on that edge, while no other vertex transform affects it. Therefore $\sigma_{\hat{i},(x,y)}^z(\{\theta_v\}) = \theta_{(x,y)}\theta_{(x,y)+\hat{i}} \, \sigma_{\hat{i},(x,y)}^z(\{+\})$. The toric code spin corresponding to a general Ising configuration is then given by

$$
\sigma_{\hat{i},(x,y)}^z(\{\theta_v\}) = \begin{cases} -\theta_{(x,y)}\theta_{(x,y)+\hat{i}}, & \text{if } x = N_x \text{ and } \hat{i} = \hat{x}, \\ \theta_{(x,y)}\theta_{(x,y)+\hat{i}}, & \text{otherwise.} \end{cases}
\tag{B.6}
$$

This allows us to write the ground state from Equation (B.3) as

$$
|GS_{-+}(\beta)\rangle = \sum_{\{\theta_v\}} e^{\beta(\sum_{x=1}^{N_x}\sum_{y=1}^{N_y}\theta_{(x,y)}\theta_{(x,y+1)} + \sum_{y=1}^{N_y}(-\theta_{(N_x,y)}\theta_{(1,y)} + \sum_{x=1}^{N_x}\theta_{(x,y)}\theta_{(x+1,y)}))/2} \prod_v A_v^{\theta_v} |\text{ref.}\rangle \,.
\tag{B.7}
$$

The norm of this state is then equal to

$$
Z_{-+}(\beta) = 2 \sum_{\{\theta_v\}} e^{\beta(\sum_{x=1}^{N_x}\sum_{y=1}^{N_y}\theta_{(x,y)}\theta_{(x,y+1)} + \sum_{y=1}^{N_y}(-\theta_{(N_x,y)}\theta_{(1,y)} + \sum_{x=1}^{N_x}\theta_{(x,y)}\theta_{(x+1,y)}))} \,,
$$

where the factor of two arises from the fact that there are two Ising configurations for each toric code configuration, and these configurations should not be treated as orthogonal. Apart from this factor of two, this norm is equal to an Ising partition function with antiperiodic-periodic boundary conditions, as given in Refs. [50] and [51]. Similarly, the norms for the even-odd and odd-odd ground states are given by the partition function for the Ising model with periodic-antiperiodic and antiperiodic-antiperiodic boundary conditions respectively. There are exact expressions for these partition functions, which are (for $\beta \geq 0$)

$$
\begin{aligned}
Z_{++}(\beta) &= C_o + S_o + C_e - S_e \,, \\
Z_{+-}(\beta) &= C_o - S_o + C_e + S_e \,, \\
Z_{-+}(\beta) &= C_o + S_o - C_e + S_e \,, \\
Z_{--}(\beta) &= -C_o + S_o + C_e + S_e \,,
\end{aligned}
$$

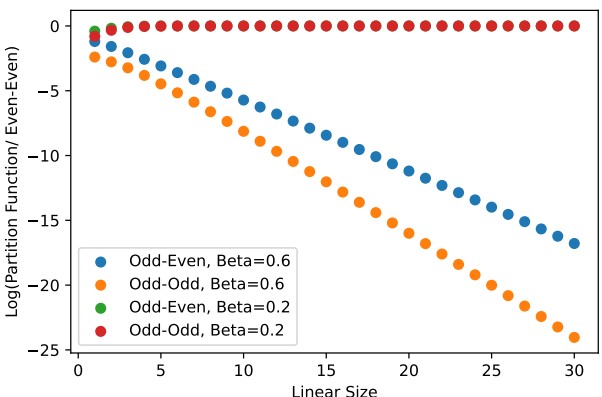

Figure 9: We plot the logarithm of the partition function ratio as a function of system size for two values of $\beta$, one below the transition and one above it. We see that the logarithm tends towards zero below the transition (as we expect from Figure 3), but drops approximately linearly with system size above the transition, implying that the partition function ratio decays exponentially with system size.

where

$$C_o = 2^{N_x N_y} \prod_{p=0}^{N_x} \prod_{q=0}^{N_y} \left[ \cosh^2(2\beta) - \cos\left(\frac{(2p+1)\pi}{N_x}\right) \sinh(2\beta) - \cos\left(\frac{(2q+1)\pi}{N_y}\right) \sinh(2\beta) \right]^{1/2},$$
(B.8)

$$S_o = 2^{N_x N_y} \prod_{p=0}^{N_x} \prod_{q=0}^{N_y} \left[ cos^2(2\beta) - \cos\left(\frac{2p\pi}{N_x}\right) \sinh(2\beta) - \cos\left(\frac{(2q+1)\pi}{N_y}\right) \sinh(2\beta) \right]^{1/2}, \quad \text{(B.9)}$$

$$C_e = 2^{N_x N_y} \prod_{p=0}^{N_x} \prod_{q=0}^{N_y} \left[ cos^2(2\beta) - \cos\left(\frac{(2p+1)\pi}{N_x}\right) \sinh(2\beta) - \cos\left(\frac{2q\pi}{N_y}\right) \sinh(2\beta) \right]^{1/2},$$
(B.10)

$$S_e = 2^{N_x N_y} \operatorname{sgn}(1 - \sinh^2(2\beta))$$
$$\times \prod_{p=0}^{N_x} \prod_{q=0}^{N_y} \left[ cos^2(2\beta) - \cos\left(\frac{2p\pi}{N_x}\right) \sinh(2\beta) - \cos\left(\frac{2q\pi}{N_y}\right) \sinh(2\beta) \right]^{1/2}. \quad \text{(B.11)}$$

As explained in Section 4.1, the action of the non-contractible deformed 't Hooft loop operators depends on ratios of these partition functions. In Figure 3 in Section 4.1, we plot these ratios for different values of $\beta$. We see that below the phase transition, the ratios are all approximately one, indicating that the deformed 't Hooft loop is acting unitarily on the ground state subspace. Above the phase transition, the ratio drops sharply towards zero (exponentially, as shown in the right side of Figure 3 where the logarithm of the ratio is plotted) indicating that the operator acts highly non-unitarily. In addition, above the phase transition the ratio of the partition function decays exponentially with system size, as shown in Figure 9, which reflects the norm-based confinement of the excitation moved by the deformed 't Hooft loop.

# C Perimeter law for 't Hooft loop

In Section 3.2, we claimed that the expectation value for the 't Hooft loop obeys a perimeter law, decaying with the length of the loop rather than the area. In this section, we will prove that result. Denoting the 't Hooft loop applied on a closed (dual) loop $c$ by $T(c)$, the expectation value is given by

$$
\begin{aligned}
\langle T(c) \rangle &= \frac{\langle GS(\beta)| T(c) |GS(\beta)\rangle}{\langle GS(\beta)|GS(\beta)\rangle} \\
&= \frac{\langle GS(0)| S(\beta)T(c)S(\beta) |GS(0)\rangle}{\langle GS(0)|S(\beta)^2|GS(0)\rangle} .
\end{aligned}
\tag{C.1}
$$

The ground state is a linear combination of closed (dual) loop configurations, weighted by the length of the loop, so we can write this expectation value as

$$
\langle T(c) \rangle = \frac{\sum_{\text{loop configs. } a} \sum_{\text{loop configs. } b} e^{-\beta L(a)} e^{-\beta L(b)} \langle a| T(c) |b\rangle}{\sum_{\text{loop configs. } a} \sum_{\text{loop configs. } b} e^{-\beta L(a)} e^{-\beta L(b)} \langle a|b\rangle} .
\tag{C.2}
$$

The operator $T(c)$ flips the edges along a dual loop $c$, meaning that there is a one-to-one mapping between loop configurations before and after the action of the operator. By defining the configuration $|T(c) : b\rangle = T(c)|b\rangle$, we can write the expectation value as

$$
\begin{aligned}
\langle T(c) \rangle &= \frac{\sum_{\text{loop configs. } a} \sum_{\text{loop configs. } b} e^{-\beta L(a)} e^{-\beta L(b)} \langle a|T(c) : b\rangle}{\sum_{\text{loop configs. } a} \sum_{\text{loop configs. } b} e^{-\beta L(a)} e^{-\beta L(b)} \langle a|b\rangle} \\
&= \frac{\sum_{\text{loop configs. } a} \sum_{\text{loop configs. } b} e^{-\beta L(a)} e^{-\beta L(b)} \delta(a, T(c) : b)}{\sum_{\text{loop configs. } a} \sum_{\text{loop configs. } b} e^{-\beta L(a)} e^{-\beta L(b)} \delta(a, b)} \\
&= \frac{\sum_{\text{loop configs. } a} e^{-\beta L(a)} e^{-\beta L(T(c):a)}}{\sum_{\text{loop configs. } a} e^{-2\beta L(a)}} .
\end{aligned}
\tag{C.3}
$$

Now, because $T(c)$ acts as a bijective mapping between loop configurations, we can write $\sum_{\text{loop configs. } a} f(a) = \sum_{\text{loop configs. } T(c):a} f(a) = \sum_{\text{loop configs. } a} f(T(c) : a)$ and so

$$
\sum_{\text{loop configs. } a} f(a) = \frac{1}{2} \sum_{\text{loop configs. } a} (f(a) + f(T(c) : a)) .
$$

Therefore

$$
\langle T(c) \rangle = \frac{\sum_{\text{loop configs. } a} e^{-\beta L(a)} e^{-\beta L(T(c):a)} + e^{-\beta L(T(c):a)} e^{-\beta L(a)}}{\sum_{\text{loop configs. } a} e^{-2\beta L(a)} + e^{-2\beta L(T(c):a)}} .
\tag{C.4}
$$

Then, writing $L(a) + L(T(c) : a) = 2\bar{L}(a)$ and $|L(a) - L(T(c) : a)| = \Delta L(a)$, we obtain

$$
\begin{aligned}
\langle T(c) \rangle &= \frac{\sum_{\text{loop configs. } a} 2e^{-2\beta \bar{L}(a)}}{\sum_{\text{loop configs. } a} e^{-2\beta \bar{L}(a)}(e^{-\beta \Delta L(a)} + e^{\beta \Delta L(a)})} \\
&= \frac{\sum_{\text{loop configs. } a} e^{-2\beta \bar{L}(a)}}{\sum_{\text{loop configs. } a} e^{-2\beta \bar{L}(a)} \cosh(\beta \Delta L(a))} .
\end{aligned}
\tag{C.5}
$$

Now consider $\cosh(\beta \Delta L(a))$. The largest $\Delta L(a)$ can be is the length $L$ of the 't Hooft loop. Because every term in the denominator is positive, this means that the denominator satisfies the inequality

$$
\sum_{\text{loop configs. } a} e^{-2\beta \bar{L}(a)} \cosh(\beta \Delta L(a)) \le e^{-2\beta \bar{L}(a)} \cosh(\beta L) .
$$

The overall expectation value therefore satisfies the inequality

$$\langle T(c) \rangle \geq \frac{\sum_{\text{loop configs. } a} e^{-2\beta \bar{L}(a)}}{\sum_{\text{loop configs. } a} e^{-2\beta \bar{L}(a)} \cosh(\beta L)} = \frac{1}{\cosh(\beta L)}. \tag{C.6}$$

This means that the expectation value of the 't Hooft loop decays at most as fast as the length of the loop for large loops ($\langle T(c) \rangle \geq e^{-|\beta|L}$). In fact, for large (positive) $\beta$ we expect that the dominant contribution to both numerator and denominator will come from configurations with few down spins near the 't Hooft loop, for which $\Delta L(a) \approx L$ and so the bound should be nearly saturated:

$$\langle T(c) \rangle \approx \frac{1}{\cosh(\beta L)}. \tag{C.7}$$

# D  Inhomogeneous deformed toric code

So far, we have considered the case where we deform the toric code equally across all of space. However, the ground states can be determined exactly even if the parameter $\beta$ varies spatially. This leads to ground states of the form

$$|GS(\{\beta_i\})\rangle \propto e^{\sum_i \beta_i \sigma_i^z / 2} |GS(0)\rangle, \tag{D.1}$$

for Hamiltonians of the form

$$H(\{\beta_i\}) = - \sum_{\text{plaquettes, } p} B_p + \sum_{\text{vertices, } v} Q_v(\{\beta_i\}), \tag{D.2}$$

where

$$Q_v(\{\beta_i\}) = e^{-\sum_{i \in \text{star}(v)} \beta_i \sigma_i^z} - \prod_{i \in \text{star}(v)} \sigma_i^x. \tag{D.3}$$

This can be further generalized by swapping some of the exponentials in the ground state expression for similar expressions involving $\sigma_i^x$:

$$|GS(\{\beta_i\}, \{\gamma_j\})\rangle \propto e^{\sum_{i \in S_1} \beta_i \sigma_i^z / 2} e^{\sum_{j \in S_2} \gamma_j \sigma_j^x / 2} |GS(0)\rangle, \tag{D.4}$$

where $S_1$ and $S_2$ are disjoint sets of edges. Note that for each edge, we apply at most one exponential term, which avoids the non-commutativity of the exponentials in $\sigma_i^z$ and $\sigma_i^x$. To build a Hamiltonian which has these ground states, we must modify both the vertex and plaquette terms of the toric code:

$$
\begin{aligned}
H(\{\beta_i\}, \{\gamma_j\}) &= \sum_{\text{plaquettes, } p} (e^{-\sum_{j \in p} \gamma_j \sigma_j^x} - \prod_{j \in p} \sigma_j^z) + \sum_{\text{vertices, } v} (e^{-\sum_{i \in \text{star}(v)} \beta_i \sigma_i^z} - \prod_{i \in \text{star}(v)} \sigma_i^x) \\
&:= \sum_{\text{plaquettes, } p} R_p(\{\gamma_j\}) + \sum_{\text{vertices, } v} Q_v(\{\beta_i\}).
\end{aligned} \tag{D.5}
$$

To verify that this Hamiltonian has ground states given by Equation (D.4), we first show that the eigenvalues of $Q_v(\{\beta_i\})$ and $R_p(\{\gamma_j\})$ are non-negative, following the approach used in Ref. [28] for the homogeneous case. We have

$$
\begin{aligned}
Q_v(\{\beta_i\})^2 &= (e^{-\sum_{i \in \text{star}(v)} \beta_i \sigma_i^z} - \prod_{i \in \text{star}(v)} \sigma_i^x)^2 \\
&= e^{-\sum_{i \in \text{star}(v)} 2\beta_i \sigma_i^z} + (\prod_{i \in \text{star}(v)} \sigma_i^x)^2 - [\prod_{i \in \text{star}(v)} \sigma_i^x] e^{-\sum_{i \in \text{star}(v)} \beta_i \sigma_i^z} - e^{-\sum_{i \in \text{star}(v)} \beta_i \sigma_i^z} \prod_{i \in \text{star}(v)} \sigma_i^x \\
&= e^{-\sum_{i \in \text{star}(v)} 2\beta_i \sigma_i^z} + 1 - (e^{-\sum_{i \in \text{star}(v)} \beta_i \sigma_i^z} + e^{+\sum_{i \in \text{star}(v)} \beta_i \sigma_i^z}) \prod_{i \in \text{star}(v)} \sigma_i^x,
\end{aligned}
$$

where we used the relation $\sigma_i^x e^{-\beta \sigma_i^z} = e^{+\beta \sigma_i^z} \sigma_i^x$. Then we can write this as

$$
\begin{aligned}
Q_v(\{\beta_i\})^2 &= (e^{-\sum_{i \in \text{star}(v)} \beta_i \sigma_i^z} + e^{+\sum_{i \in \text{star}(v)} \beta_i \sigma_i^z}) e^{-\sum_{i \in \text{star}(v)} \beta_i \sigma_i^z} \\
&\quad - (e^{-\sum_{i \in \text{star}(v)} \beta_i \sigma_i^z} + e^{+\sum_{i \in \text{star}(v)} \beta_i \sigma_i^z}) \prod_{i \in \text{star}(v)} \sigma_i^x \\
&= 2\cosh(\sum_{i \in \text{star}(v)} \beta_i \sigma_i^z)(e^{-\sum_{i \in \text{star}(v)} \beta_i \sigma_i^z} - \prod_{i \in \text{star}(v)} \sigma_i^x) \\
&= 2\cosh(\sum_{i \in \text{star}(v)} \beta_i \sigma_i^z) Q_v(\{\beta_i\}).
\end{aligned}
$$

For any eigenstate $|\psi\rangle$ of $Q_v(\{\beta_i\})$, with eigenvalue $\lambda$, we therefore have

$$
Q_v(\{\beta_i\})^2 |\psi\rangle = \lambda^2 |\psi\rangle = 2\cosh(\sum_{i \in \text{star}(v)} \beta_i \sigma_i^z) \lambda |\psi\rangle .
$$

This implies that either $\lambda = 0$ or $\lambda |\psi\rangle = 2\cosh(\sum_{i \in \text{star}(v)} \beta_i \sigma_i^z) |\psi\rangle$, meaning that $Q_v(\{\beta_i\})$ shares its non-zero eigenvalues with $2\cosh(\sum_{i \in \text{star}(v)} \beta_i \sigma_i^z)$. Because the eigenvalues of $\cosh(\sum_{i \in \text{star}(v)} \beta_i \sigma_i^z)$ are all positive for real $\beta$, this means that the eigenvalues of $Q_v(\{\beta_i\})$ are non-negative. A similar result holds for $R_p(\{\gamma_j\})$ which has the same algebraic structure (but with $\sigma^z$ and $\sigma^x$ swapped). As a result, if we find a state which has $Q_v(\{\beta_i\}) = 0$ for all $v$ and $R_p(\{\gamma_j\}) = 0$ for all $p$ then it is an eigenstate of each energy term with minimum energy and so is a ground state.

Now we can verify that the claimed ground states from Equation (D.4) satisfy these conditions. We have

$$
\begin{aligned}
&Q_v(\{\beta_i\}) |GS(\{\beta_i\}, \{\gamma_j\})\rangle \\
&\quad \propto Q_v(\{\beta_i\}) e^{\sum_{i \in S_1} \beta_i \sigma_i^z/2} e^{\sum_{j \in S_2} \gamma_j \sigma_j^x/2} |GS(0)\rangle \\
&\quad = (e^{-\sum_{i \in \text{star}(v)} \beta_i \sigma_i^z} - \prod_{i \in \text{star}(v)} \sigma_i^x) e^{\sum_{i \in S_1} \beta_i \sigma_i^z/2} e^{\sum_{j \in S_2} \gamma_j \sigma_j^x/2} |GS(0)\rangle \\
&\quad = (e^{-\sum_{i \in \text{star}(v)} \beta_i \sigma_i^z} - \prod_{i \in \text{star}(v)} \sigma_i^x) e^{\sum_{j \in \text{star}(v)} \beta_j \sigma_j^z/2} e^{\sum_{i \in S_1 \notin \text{star}(v)} \beta_i \sigma_i^z/2} e^{\sum_{j \in S_2} \gamma_j \sigma_j^x/2} |GS(0)\rangle \\
&\quad = (e^{-\sum_{i \in \text{star}(v)} \beta_i \sigma_i^z} e^{\sum_{j \in \text{star}(v)} \beta_j \sigma_j^z/2} - \prod_{i \in \text{star}(v)} \sigma_i^x e^{\sum_{j \in \text{star}(v)} \beta_j \sigma_j^z/2}) \\
&\qquad \times e^{\sum_{i \in S_1 \notin \text{star}(v)} \beta_i \sigma_i^z/2} e^{\sum_{j \in S_2} \gamma_j \sigma_j^x/2} |GS(0)\rangle \\
&\quad = (e^{-\sum_{i \in \text{star}(v)} \beta_i \sigma_i^z/2} - e^{-\sum_{j \in \text{star}(v)} \beta_j \sigma_j^z/2} \prod_{i \in \text{star}(v)} \sigma_i^x) \\
&\qquad \times e^{\sum_{i \in S_1 \notin \text{star}(v)} \beta_i \sigma_i^z/2} e^{\sum_{j \in S_2} \gamma_j \sigma_j^x/2} |GS(0)\rangle \\
&\quad = e^{-\sum_{i \in \text{star}(v)} \beta_i \sigma_i^z/2} (1 - \prod_{i \in \text{star}(v)} \sigma_i^x) e^{\sum_{i \in S_1 \notin \text{star}(v)} \beta_i \sigma_i^z/2} e^{\sum_{j \in S_2} \gamma_j \sigma_j^x/2} |GS(0)\rangle \\
&\quad = e^{-\sum_{i \in \text{star}(v)} \beta_i \sigma_i^z/2} e^{\sum_{i \in S_1 \notin \text{star}(v)} \beta_i \sigma_i^z/2} e^{\sum_{j \in S_2} \gamma_j \sigma_j^x/2} (1 - \prod_{i \in \text{star}(v)} \sigma_i^x) |GS(0)\rangle .
\end{aligned}
$$

Then we note that $\prod_{i \in \text{star}(v)} \sigma_i^x$ acts as the identity on any toric code ground state $|GS(0)\rangle$ and so

$$
(1 - \prod_{i \in \text{star}(v)} \sigma_i^x) |GS(0)\rangle = 0.
$$

This means that

$$
Q_v(\{\beta_i\}) |GS(\{\beta_i\}, \{\gamma_j\})\rangle = 0,
$$

for all vertices $v$. Following exactly the same reasoning (and noting that $S_1$ and $S_2$ are disjoint sets so the two exponential factors in the ground state expression commute), we also have $B_p |GS(\{\beta_i\}, \{\gamma_j\})\rangle = 0$ for all plaquettes $p$, indicating that $|GS(\{\beta_i\}, \{\gamma_j\})\rangle$ is indeed a ground state.

Now consider how the original toric code ribbon operators act in this new model. Due to the exponential factors involving both $\sigma^x$ and $\sigma^z$, neither the electric nor magnetic ribbon operators will produce isolated excitations at the ends of the ribbons. However, just as we did for the magnetic ribbon operator in the homogeneous case, we can define deformed ribbon operators that only produce excitations at the ends of the ribbons when acting on the ground state, with the caveat that these operators are non-unitary. Defining

$$S(\{\beta_i\}) = \prod_{i \in S_1} e^{\beta_i \sigma_i^z/2}, \tag{D.6}$$

and

$$R(\{\gamma_i\}) = \prod_{i \in S_2} e^{\gamma_i \sigma_i^x/2}, \tag{D.7}$$

we can write the deformed magnetic ribbon operator as

$$\tilde{C}(t) = S(\{\beta_i\}) C(t) S(\{\beta_i\})^{-1}, \tag{D.8}$$

and the deformed electric ribbon operator as

$$\tilde{L}(t) = R(\{\gamma_i\}) L(t) R(\{\gamma_i\})^{-1}. \tag{D.9}$$

In terms of local operators, these ribbon operators can be written as

$$\tilde{C}(t) = \prod_{i \in t} e^{\beta_i \sigma_i^z/2} \sigma_i^x e^{-\beta_i \sigma_i^z/2}, \tag{D.10}$$

and

$$\tilde{L}(t) = \prod_{i \in t} e^{\gamma_i \sigma_i^x/2} \sigma_i^z e^{-\gamma_i \sigma_i^x/2}, \tag{D.11}$$

where we define $\beta_i = 0$ for $i$ outside of $S_1$ and similarly $\gamma_i = 0$ for $i$ outside of $S_2$. Both of these ribbon operators are topological and so the closed versions of these operators have expectation values that satisfy a zero-law, being independent of length or area. However, the open ribbon operators do not necessarily produce energy eigenstates and are expected to be poor descriptions of the basic excitations when the $\gamma$ or $\beta$ variables are large (due to condensation).

It is interesting that we can construct exact ground states even with a spatially varying parameter, especially because we can tune this parameter over space so that it crosses a phase transition. The fact that an excitation can be confined in one region of space and unconfined in another could potentially be used to trap excitations and to braid them in a predictable way. However, we note that the form of the Hamiltonian seems to be rather fine tuned and the ground state properties at large $\beta_i$ and $\gamma_i$ are unlikely to be robust as they are in the topological phase.

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
