# Peer review of "Gaining insights on anyon condensation and 1-form symmetry breaking across a topological phase transition in a deformed toric code model"

_SciPost Physics, doi:SciPost Phys. 15, 253 (2023)_

## Round 2 · Referee Report · Anonymous · 2023-7-3

Report
This manuscript provides an interesting viewpoint on topological phase transition(confinement --deconfinement) that is not driven by anyon condensate, and the wilson line operator persists to act like perimeter law. In particular, the universality is akin to 2d classical Ising while the conventional confinement --deconfinement transition is 2+1s quantum Ising.
I have a few questions:
1) for conventional confinement --deconfinement transition of toric code, one can demonstrate that the confinement --Higgs has no transition and is smoothly connected. what is the case here?
2) As the PEPES wave function during the transition can be mapped into the partition function of classical ising at finite T, can we demonstrate the area law of wilson line(X) by looking into the property of the classical Ising ? (I think the X wilson line will be mapped into the charge operator inside?)
3) the conclusion of this paper should be generalized into other CSS code(discrete gauge theory). One can introduce the `beta' component to any PEPS wavefunction of CSS code and that can potentially drive the deconfinement-confinement transition. the wilson operator will persist to be perimeter law for such transitions?
4) The scheme of the transition, as well as the perimeter law of the wilson operator is also mentioned here https://arxiv.org/abs/2301.05687, although they are looking into the decoherence transition.
Author: Joe Huxford on 2023-11-29 [id 4162]
(in reply to Report 1 on 2023-07-03)
Dear Referee,
Thank you again for the valuable comments that helped improve our manuscript. Although we didn't receive the requested clarification about these comments, we have attempted to address them as best we can. We will shortly resubmit the paper, so we felt it would be best to explain how we have addressed your comments here.
First Referee comment 1): For conventional confinement--deconfinement transition of toric code, one can demonstrate that the confinement--Higgs has no transition and is smoothly connected. what is the case here?
As we mentioned in our prior response, the model does not directly address the transition to the Higgs phase and so we felt it would not be appropriate to comment on this in the paper. However, as the model realises the same confinement phase as for the linear magnetic field, it should be smoothly connected to the Higgs phase in the same way.
First Referee comment 2): As the PEPES wave function during the transition can be mapped into the partition function of classical ising at finite T, can we demonstrate the area law of wilson line(X) by looking into the property of the classical Ising ? (I think the X wilson line will be mapped into the charge operator inside?).
We took wilson line (X) here to refer to the 't Hooft loop, which is the product of $\sigma_i^x$ operators along a dual loop. In the paper, we demonstrated that it obeys a perimeter law. However the referee suggests that this could be examined using the classical Ising model. We examined this possibility and found that, while this is possible, the calculation is similar to the one we performed in Appendix C. We have attached a document (HooftLoopFromIsing) showing our workings on this front. Specifically, even though the 't Hooft loop is mapped to a product of vertex terms in the region enclosed by the dual loop, as the referee suggests, when evaluating its expectation value in the Ising basis it reduces to the expectation value of a quantity on the boundary of the region in the classical Ising model. We could then use the Ising expectation value to put limits on the 't Hooft loop expectation value, but it is equivalent to the expression that we found in Appendix C so we decided not to include this in the paper.
First Referee comment 3): The conclusion of this paper should be generalized into other CSS code(discrete gauge theory). One can introduce the beta component to any PEPS wavefunction of CSS code and that can potentially drive the deconfinement-confinement transition. the wilson operator will persist to be perimeter law for such transitions?
We have included a new section in the main text, Section 5, covering how more general commuting projector models can be "deformed" in a similar way to the deformed toric code. We also considered the specific case of quantum double models (discrete lattice gauge theory) and demonstrated that the 't Hooft loops still satisfy the perimeter law (the objects that we call Wilson loops satisfy a zero law).
First Referee comment 4): }The scheme of the transition, as well as the perimeter law of the wilson operator is also mentioned here https://arxiv.org/abs/2301.05687, although they are looking into the decoherence transition.
Thank you for bringing this interesting paper to our attention, but we felt it was sufficiently removed from our work that a citation was not necessary. As the referee says, the authors of that work are looking into the decoherence transition instead and we could not determine the exact relevance to our work in the absence of clarification.
Yours Sincerely,
Authors
Attachment:
Author: Joe Huxford on 2023-07-12 [id 3805]
(in reply to Report 1 on 2023-07-03)
Dear Referee,
Thank you for your report on our work. You raise a number of interesting questions, but before we respond in full we would like to ask for clarification on some of those points.
1) You ask about the connection between the confined and Higgs phase, pointing out that these are normally smoothly connected phases (at least, there is no bulk transition). When you ask "what is the case here?", do you mean whether those same phases can be smoothly connected while staying within the model? We only consider the toric code phase and confined phase (although the generalized version in the appendix can be in the Higgs phase), and so the model does not describe the transition between the Higgs and confined phases.
2) By the area law of Wilson line X, do you mean what we call the 't Hooft loop (i.e. a string of X operators along a dual path)? In that case we actually find a perimeter law. We believe you can express the expectation value of a closed loop in terms of Ising variables, but it would be complicated to evaluate that expression because it would involve a product across an area.
3) For the CSS codes, we are unsure how to treat generic CSS codes in this scheme. If you just want to examine discrete gauge theories, we think that these could be treated with this scheme, but it may be beyond the scope of this paper.
4) Thank you for bringing the linked paper to our attention. It's an interesting paper, but as you say they consider decoherence and mixed states. Is there a specific part of the paper that you think we should address?
Yours sincerely, Authors
Author: Joe Huxford on 2023-11-29 [id 4161]
(in reply to Report 2 on 2023-10-13)Dear Referee,
Thank you for your careful reading of our manuscript and for constructive criticism, which have helped us to improve our work. We will shortly resubmit the paper with appropriate modifications, but we also give a more detailed response to your questions here, in a comment-response format.
Second Referee comment 1): The deformed 't Hooft operator is non-unitary. Under the framework of using higher-form symmetry to characterize topological order, is there any caveat to use non-unitary order parameter for the 1-form symmetry?
In the context of our paper, the answer has two parts due to the two roles of the deformed 't Hooft loop:
The first part is the role of the deformed ‘t Hooft loop in the spontaneous breaking of the 1-form electric symmetry. There is no caveat to using a non-unitary operator as the order parameter. The use of the non-unitary operator is to show that there are states in the ground state manifold that have different charges under the $\mathbb{Z}_2$ 1-form electric symmetry. As we’ve shown in our paper, the non-contractible deformed ‘t Hooft loop maps between states in the ground state manifold. The non-zero expectation value of the non-contractible deformed ‘t Hooft loop on the ground state manifold implies that the 1-form electric symmetry is broken.
The second part is the use of the non-unitary deformed ‘t Hooft loop as an approximate 1-form symmetry to characterize topological order. The deformed ‘t Hooft loop operator is topological, in that we can smoothly change the geometry of the loop without changing its operation. However, due to its non-unitary nature, the deformed ‘t Hooft loop does not preserve the norm. As we’ve shown in Section 4 and also in Appendix B, in the topological phase, the deformed ‘t Hooft loops are approximately unitary since the operator almost preserves the norm when acting on states in the ground state manifold. In this sense, the deformed ‘t Hooft loop can be treated as an approximate emergent 1-form magnetic symmetry operator. We still have the emergent mixed ‘t Hooft anomaly between electric and magnetic 1-form symmetries, which is an indicator of the topological phase. In the trivial phase, the deformed ‘t Hooft operator is no longer approximately unitary and so should not be treated as a symmetry, meaning that there is no emergent mixed ‘t Hooft anomaly between 1-form symmetries. Furthermore, the ground states are now distinguishable, as discussed in Section 4.3 in the manuscript (and the response for the below comment).
In conclusion, the order parameter is just required to transform non-trivially under the symmetry, so it does not need to be unitary. We have added a comment to this effect in Section 4.2 (before Equation 4.10). We also removed a comment at the end of Section 4.2 about the non-unitary nature not being ``ideal" which may have been confusing.}
Second Referee comment 2): How to characterize this topological to trivial phases transition if the 1-form symmetry breaks on both sides and the degeneracy is not lifted.
We believe that the character of the phase transition is still the usual condensation-confinement transition and this is the best way to think of it, despite the norm-based confinement of the magnetic excitations. We have added comments to the conclusion to emphasise this point. If we do want to use higher-form symmetry to characterize the phase transition, then we could differentiate between having two approximate symmetries in the topological phase, compared to just one symmetry in the trivial phase. As we discuss in Section 4.1 (and prove in Appendix B), the deformed 't Hooft loop is approximately unitary in the topological phase, so it acts like a true higher-form symmetry, whereas in the trivial phase it is not (although, as we mentioned in the response to the referee's previous question, it is still a good order parameter, just not a symmetry).
We provided the criteria to differentiate the topological phase and trivial phase in Section 4.3 of our manuscript. The ground state degeneracy is not lifted because the spontaneous breaking of the 1-form electric symmetry survives on both sides of the topological phase transition. However, in the topological phase, the ground states are indistinguishable by any local operator. In the trivial phase, the ground states are distinguishable by a local operator, which can be understood by the non-unitary nature of the deformed ‘t Hooft loop. A local operator can have different expectation values on the two ground states that are connected by a non-contractible deformed ‘t Hooft loop. One can see this explicitly by the following calculation. Let us consider $|GS2\rangle =\tilde{T}(c)|GS1\rangle$, where $\tilde{T}(c)$ is a non-contractible deformed ‘t Hooft loop. \begin{align} \langle GS2|O|GS2\rangle = \langle GS1|\tilde{T}^\dagger(c)O \tilde{T}(c)|GS1\rangle \nonumber =\langle GS1|\tilde{T}^\dagger(c’) \tilde{T}(c’)O|GS1\rangle \nonumber
\end{align} Here $c’$ is a smooth deformation of $c$ such that $c’$ does not intersect with the local operator $O$. In the topological phase, $\tilde{T}(c’)$ is approximately unitary when acting on a ground state, so we conclude that $\langle GS2|O|GS2\rangle \approx \langle GS1|O|GS1\rangle$ since $\tilde{T}^\dagger(c’) \tilde{T}(c’) \approx 1$. This approximate equality between expectation values is exact in the fixed point (where the deformed 't Hooft loop is the regular 't Hooft loop) and is robust in the thermodynamic limit (from the properties of the topological phase). In the trivial phase, where $\tilde{T}(c’)$ is not approximately unitary, one can find a local operator $O$ such that $\langle GS2|O|GS2\rangle$ and $\langle GS1|O|GS1\rangle$ differ significantly.}
Second Referee comment 3): If a perturbation (e.g. a field in the z-direction) is added to the deformed Toric Code Hamiltonian, will the expectation value of deformed magnetic ribbon operator become area law in the trivial phase?
Although we do not have a detailed calculation, if a large enough perturbation was added, then we expect that the expectation value would become area law. This does occur for the undeformed magnetic ribbon operator for a pure linear field term (see Ref. 41 by M. B. Hastings and X.-G. Wen). We expect that the exponential term in the Hamiltonian, as well as the exponential term in the deformed ribbon operator, should only further decrease the expectation value of the ribbon operator. However we do not expect an infinitesimal perturbation to produce the area law. }
Second Referee comment 4): It is not trivial to see Eq. 2.4 is the ground state of the deformed Toric Code Hamiltonian.
We have added a comment to the paper pointing out that this was demonstrated by C. Castelnovo and C. Chamon in their paper (Ref. 28 in our paper), and also pointing to Appendix D, where we demonstrate it for a more general model. The proof involves showing that the deformed terms have zero as their lowest eigenvalue and then showing that they annihilate the putative ground state.
Yours Sincerely, Authors

---

## Round 2 · Referee Report · Anonymous · 2023-10-13

Report
This paper discusses a deformed Toric Code model where the spontaneous 1-form symmetry breaking is not sufficient to characterize the topological phase as the trivial phase of the model also exhibits 1-form symmetry breaking. It brings about an interesting question if 1-form symmetry breaking is a necessary and sufficient condition for the topological order. Regarding that, I have couples of questions that I want to clarify.
1. The deformed 't Hooft operator is non-unitary. Under the framework of using higher-form symmetry to characterize topological order, is there any caveat to use non-unitary order parameter for the 1-form symmetry?
2. How to characterize this topological to trivial phases transition if the 1-form symmetry breaks on both sides and the degeneracy is not lifted.
3. If a perturbation (e.g. a field in the z-direction) is added to the deformed Toric Code Hamiltonian, will the expectation value of deformed magnetic ribbon operator become area law in the trivial phase?
3. It is not trivial to see Eq. 2.4 is the ground state of the deform Toric Code Hamiltonian.

---

## Round 3 · Referee Report · Anonymous (Referee 1) · 2023-12-4

Report

The revised version addressed my concerns and is good for publication.

---

## Round 3 · Author Response

Dear Editor,

We would like to resubmit a revised version of the manuscript to Scipost Physics. We thank the referees for their useful comments and are encouraged by the overall positive feedback our manuscript received. We have addressed the referees’ concerns in detail in responses to their reports and made corresponding adjustments to the manuscript. We believe that after these revisions, we have answered all of the referees’ queries, and that our work is now appropriate for publication in SciPost Physics.

Sincerely,
Joe Huxford, on behalf of the authors

---

## Round 3 · List of Changes

We made the following changes to the paper:

We added a comment in Section 4 to clarify that the deformed 't Hooft loop, which plays the role of order parameter, does not need to be unitary.

We added a new Section 5 to generalize our formalism to many different commuting projector models, as well as different types of deformation. In particular, we construct a deformed version of the quantum double model and show how the more general formalism can reproduce the deformed toric code.

We added a comment above equation (2.4) to point out the derivation in Appendix D.

We added some additional comments in the conclusion section to clarify the character of the phase transition and to summarize the new section (Section 5).

We also made assorted minor changes to make mathematical notation clearer.

---

## Editorial Decision

published